# Denoising Efficiency and Lines Matching Models

## Abstract

In this paper, we analyze the denoising loss used by key denoising models and identify an inefficiency that stems from the random pairing which they employ between samples from the source and target distributions. Regressing the denoiser under these non-deterministic conditions causes its predictions to collapse toward the mean of the source or target distributions. We show that this degeneracy creates false basins of attraction, distorting the denoising trajectories and ultimately increasing the number of steps required to sample these models.

We also analyze the alternative strategy of deriving the pairing from an Optimal Transport between the two distributions, and show that while this approach can alleviate this degeneracy, it suffers from a curse of dimensionality, where the pairing set size must scale exponentially with the signal's dimension.

In order to empirically validate and utilize these theoretical observations, we design a new training approach that circumvents these pitfalls by leveraging the deterministic ODE-based samplers, offered by certain denoising diffusion and score-matching models. These deterministic samplers establish a well-defined change-of-variables between the source and target distributions. We use this correspondence to construct a new probability flow model, the *Lines Matching Model* (LMM), which matches globally straight lines interpolating between the two distributions. We show that the flow fields produced by the LMM exhibit notable temporal consistency, resulting in trajectories with excellent straightness scores, and allow us to exceed the quality of distilling the input correspondence.

The LMM flow formulation allows us to further enhance the fidelity of the generated samples beyond the input correspondence by integrating domain-specific reconstruction and adversarial losses. Overall, the LMM achieves state-of-the-art FID scores with minimal NFEs on established benchmark datasets: 1.57/1.39 (NFE=1/2) on CIFAR-10, 1.47/1.17 on ImageNet 64×64, and 2.68/1.54 on AFHQ 64×64.

## 1 Introduction

Diffusion models are the core engine behind many recent state-of-the-art generative models across various domains, e.g., image generation (Song et al., 2021b; Ho et al., 2020; Dhariwal & Nichol, 2021; Rombach et al., 2022), text-to-image generation (Nichol et al., 2022; Ramesh et al., 2022; Saharia et al., 2024), audio synthesis (Kong et al., 2021; Kim et al., 2021; Chen et al., 2020; Popov et al., 2021), and video generation (Ho et al., 2022; Singer et al., 2023; Liu et al., 2024b).

This gain in popularity of the underlying denoising diffusion (Sohl-Dickstein et al., 2015; Ho et al., 2020) and score-matching (Song et al., 2019; Song & Ermon, 2020; 2019) models over GANs (Goodfellow et al., 2014) is often attributed to their improved distribution reproduction (Dhariwal & Nichol, 2021), and immunity to various optimization hurdles that plague GAN training (mode collapse and forgetting (Thanh-Tung & Tran, 2020)). Nevertheless, unlike the single-step sampling of GAN and VAE (Kingma & Welling, 2014) models, the noise removal process follows non-trivial probability flow trajectories, requiring fine quadrature steps and resulting in non-negligible computational effort during inference. This ranges between hundreds of sampling steps in early methods (Ho et al., 2020) and tens in more recent ones (Karras et al., 2022).

Distilling pre-trained denoising models allows reducing the Number of Function Evaluations (NFEs) during sampling. This approach can be carried out in different ways; learning the end-to-end sampling operator (Luhman & Luhman, 2021), or reducing its number of steps progressively (Salimans & Ho, 2022) even during teacher training (Frans et al., 2025). More recently, the denoising trajectories are learned either by ensuring consistency along successive steps (Song et al., 2023), or along arbitrary segments (Kim et al., 2024). These methods offer a significant speedup over their teacher models, nevertheless, they also inherit inefficiencies inherent to the trajectories that they replicate.

As an alternative, the probability flow matching techniques in (Lipman et al., 2023; Albergo & Vanden-Eijnden, 2023b) incorporate Optimal Transport (OT) considerations in order to produce more constant flow trajectories, requiring fewer sampling steps. Additional improvement in straightness is achieved by an iterative rectification scheme in (Liu et al., 2023; 2024a), as well as by replacing the random pairing between the source and data examples with an OT pairing (Pooladian et al., 2023; Tong et al., 2024). While improving upon traditional denoising losses, the flow fields produced by these approaches still contain false attraction basins, causing the trajectories to curve.

This paper primarily aims to understand the root of the sampling inefficiency of modern diffusion models. As a first contribution, we provide an analysis showing that the ambiguous pairing between latent source noise and target data samples leads to an ill-posed regression problem, compromising the performance of key denoising models, including denoising diffusion, score and flow-matching. At low signal-to-noise ratios, this indeterminacy in the denoising loss becomes worse and causes the denoiser's predictions to collapse toward the mean of either the source or target distributions. We show that this acts as a false basin of attraction that curves the denoising trajectories, ultimately increases the number of steps needed for sampling.

We make a second theoretical contribution, where we formally show that while the OT-based pairing in (Pooladian et al., 2023; Tong et al., 2024) is a valid approach for reducing this indeterminacy and the attraction to the false basins, due to a fundamental course-of-dimensionality, the batch size required scales exponentially as a function of the signal dimension. Given that the latter is fairly high in various practical scenarios and the former is typically constrained by memory and compute limitations, the effectiveness of this approach is limited, as demonstrated in Table 1. To the best of our knowledge, this result is not reported in the diffusion models literature.

We leverage the fact that certain denoising diffusion (Song et al., 2021a) and score-matching (Song & Ermon, 2019; Karras et al., 2022) models define *deterministic* ODE-based flows, which also induce correspondence between source and target distributions for several purposes: (i) validate the sufficiency of our theoretical insights, i.e., evaluate the extent of inefficiency that the random pairing inflicts, (ii) assess whether it is better to model the induced correspondence or model their flow, and (iii) to explore how to achieve the latter while avoiding the inefficient features in the flows.

This study leads us to a new distillation strategy: unlike existing approaches that learn the full (and potentially inefficient) probability flow field, we utilize only the induced correspondence between distributions. To this end, we introduce a new probability flow model, the Lines Matching Model (LMM), trained to match globally straight lines interpolating between the distributions. As demonstrated in Table 1, the flow fields produced by LMM exhibit strong temporal consistency, resulting in trajectories with excellent straightness scores.

Beyond its sampling efficiency, the LMM addresses the distributional drift in existing self-distilling flow models, e.g., Liu et al. (2023), by incorporating domain-specific reconstruction and adversarial losses, allowing it to exceed the sampling fidelity set by its input correspondence. Overall, the LMM achieves state-of-the-art Fréchet Inception Distance (FID) scores using minimal NFEs on established benchmarks, specifically, 1.57/1.39 (NFE=1/2) for CIFAR-10, 1.47/1.17 for ImageNet 64×64, and 2.8/1.61 for AFHQ 64×64.

While outlined in the paper, the two core theoretical contributions are presented in full in Appendix A and B, and we encourage readers to consult these sections for further details.

## 2 Denoising Efficiency Analysis

Our first investigation aims to shed light on the implications of minimizing the denoising loss over independently sampled pairs of source (noisy) and data points. Prior work Pooladian et al. (2023) argues that this practice leads to nonzero gradient variance at convergence, slower training, and degraded straightness of the resulting probability paths. However, these claims are either shown empirically, primarily through suboptimal performance at low NFE, or under theoretical scenarios (infinite batch size in OT-based pairing). We complement this with an analysis offering a mechanistic explanation of how this loss, in its general use case, drives the trajectories toward singular basins of attraction, and thus undermines their sampling efficiency.

We begin by reviewing several key denoising-based generative models, with an intent to bring them to a common ground in order to highlight the source of a sampling inefficiency that they share. The Denoising Diffusion Probability Models (DDPM) (Sohl-Dickstein et al., 2015; Ho et al., 2020), as well as Denoising Score Matching (DSM) approaches, specifically the Noise Conditional Score Network (NCSN) (Song & Ermon, 2019) minimize the following form of denoising loss,

$$\mathcal{L}_{\text{denoise}} = \mathbb{E}_{t,q(x_1),p(x|x_1,t)}\Big[\|N_\theta(x,s_t) - \nabla_x \log p(x|x_1,t)\|^2\Big], \tag{1}$$

where $q(x_1)$ is the target data distribution which we are given empirically. In case of DDPM, $p(x|x_1,t) = \mathcal{N}(\sqrt{\alpha_t}x_1, (1-\alpha_t)I)$ and $s_t = t$, where $1 \le t \le T$ is a noise scheduling index weighted by probabilities $\propto (1-\alpha_t)$, and $\alpha_t = \prod_{i=1}^{t}(1-\beta_i)$ and $0 < \beta_i < 1$ are a pre-defined sequence of noise scales[1]. In this framework the network $N_\theta$ models the mean of the reverse Gaussian kernels by $p(x^{t-1}|x^t) = \mathcal{N}((x^t + \beta_t N_\theta(x^t,t))/\sqrt{1-\beta_t}, \beta_t I)$, which are designed to start their operation from a source distribution, $x^T \sim p_0 = \mathcal{N}(0,I)$. In the NCSN, $p(x|x_1,t) = \mathcal{N}(x_1, \sigma_t^2 I)$ and $s_t = \sigma_t$, where $\{\sigma_t\}_{t=1}^{T}$ are positive noise scales, weighted $\propto \sigma_t^2$. In this approach, the network $N_\theta$ models the score field of noised data densities $p(x, \sigma_t) = q * \mathcal{N}(0, \sigma_t^2 I)$, which is used for gradually denoising samples, starting from $x^T \sim p_0 = \mathcal{N}(0, \sigma_T^2 I)$, where $\sigma_T^2 >> \mathbb{V}[x_1]$. Much has been discussed about the close relation of these approaches (Vincent, 2011; Song et al., 2021b; Karras et al., 2022). This formalism can be further generalized to cover continuous-time Stochastic Diffusion Equations (SDEs), where the DDPM results in a Variance Preserving (VP) process, and the NCSN in a Variance Exploding (VE) process, see (Song et al., 2021b). To highlight the shared properties of DDPM and DSM, we adopt the unified notation $p(\cdot|\cdot)$ to represent a general conditional probability density

The Conditional Flow Matching (CFM) method in (Lipman et al., 2023), constructs conditional maps $\psi_{x_1}(x,t) = (1-t)x + tx_1$ that map $p_0$ (a normalized Gaussian), towards a small Gaussian[2] centered around each $x_1$ as a function of $t \in [0,1]$. The flow fields induced by these maps, $\partial \psi_{x_1}/\partial t$, are shown to result in straight paths and map the endpoint distributions correctly. However, in order to extend this flow to the full data distribution $q$, the network $N$ is trained to match an aggregated velocity flow field by marginalizing over all the data points $x_1$ by solving,

$$\text{argmin}_\theta \mathbb{E}_{t,q(x_1),p(x_0)}\Big[\|N_\theta(tx_1 + (1-t)x_0, t) - (x_1 - x_0)\|^2\Big], \tag{2}$$

which appears sufficiency close to Eq. 1 for our purpose.

The following proposition simplifies the denoising losses, in Eq. 1 and Eq. 2, and allows us to understand how they shapes the sampling trajectories in the learned model $N_\theta$. At $t = T$ in Eq. 1 (and $t = 0$ in Eq. 2) the noised-to-clean signal problem solved in these equations boils down to the following regression problem,

$$\text{argmin}_\theta \mathbb{E}_{q(x_1),x_0 \sim p_0}\Big[\|N_\theta(x_0, \sigma_t) - \nabla_x \log p(x_0|x_1,T)\|^2\Big], \tag{3}$$

with $p(x_0) \propto e^{-\|x_0-x_1\|^2/2}$ in case of Eq. 2. The solution to this problem is given by

$$N_\theta(x_0, \sigma_T) = \mathbb{E}_{q(x_1)}\big[\nabla_x \log p(x_0|x_1,T)\big]. \tag{4}$$

---

[1]The $\alpha_t$ defined here correspond to the $\bar{\alpha}_t$ in the derivation of Ho et al. (2020).
[2]To simplify derivation we assume a zero width target Gaussian around each data point, i.e., $\sigma_{\min} = 0$ in the formalism of (Lipman et al., 2023)

We provide the derivation of this proposition and the explicit calculation of $\nabla_x \log p$ for DDPM, NCSN, and CFM in Appendix A. While this analysis applies to the very early stage of the flow, it is rather consequential, as we discuss next.

**Implications.** The noised-to-clean signal regression problem solved in Eq. 3 is fairly general and is known to *underestimate* the true regression (Kendall & Stuart, 1973; Clarke & Gorder, 2013), due to averaging caused by the noise present in $p(x|x_1, t)$ at $t = T$ (and $t = 0$ in CFM). Indeed, at this extreme limit of low Signal-to-Noise Ratio (SNR), where $p(x|x_1, t) \approx p_0(x)$, the source and target samples are randomly paired in Eq. 3. As we show in Appendix A, the regressed model in Eq. 4 collapses to a constant prediction, specifically, $N(x, T) \approx \mathbb{E}_{p_0}[x] = \mu_0 - x$ in the case of DDPM, and $N(x, \sigma_T) \approx \mathbb{E}_q[x_1] \propto \mu_1 - x$ in the cases of NCSN and CFM.

This implies that the randomly-paired denoising loss, used by these models, creates flows that start by uniformly directing all the source samples $x_0$ towards a fixed point, $\mu_0$ in the case of DDPM and $\mu_1$ in the NCSN and CFM, rather than towards diverse instances $x_1$ in the target distribution $q$. This effect is highlighted in Table 1 by the green arrows. At later times $t$, which correspond to higher SNR, the trajectories recover from the effect of these false basins, and redirect towards specific samples. This shift in direction, also visible in the paths shown in Table 1, is the reason behind the undermined straightness that reduces the sampling efficiency by requiring finer integration steps to accurately trace the resulting curved trajectories. Indeed, replacing this random pairing with a deterministic one, as done in LMM (described below) and 2-RectFlow, results in a dramatic straightening of the trajectories.

In this context, let us mention several related works. An alternative derivation to the CFM in (Albergo & Vanden-Eijnden, 2023a) discusses the option of optimizing the transport of their maps and proposes an initial direction to shorten their path length. Training this model to take larger step sizes results in improved performance at lower NFEs in Frans et al. (2025).

The flow rectification process described in (Liu et al., 2023) also matches the flow using Eq. 2, however it operates iteratively; at each step $k$ it trains $N^k$ over a different set of source $Z_0^k$ and target $Z_1^k$ examples. The process starts with the *random* pairing at $k = 0$, but in the following steps, $Z_0^{k+1}$ and $Z_1^{k+1}$ are produced by generating new samples using $N^k$ starting from $p_0$ and $q$ (by integrating $-N^k$). This results in a *deterministic* pairing and this process is shown to monotonically increase the straightness of the trajectories in $N^k$. It is shown in (Wang et al., 2025) that extending the loss in Eq. 2, from lines (between $x_0$ and $x_1$) to a wider class of first-order paths, improves the training efficiency and performance.

As shown in Table 1, the resulting flow trajectories at $k\!=\!1$ share a similar gravitation towards $\mu_1$ as in the CFM. At $k\!=\!2$ they become significantly more straight and easier to integrate. However, as $k$ increases errors in the estimated flow field $N^k$ accumulate and cause $Z_0^k$ and $Z_1^k$ to drift from $p_0$ and $q$ respectively. 2-Rect-Flow ($k\!=\!2$) is said to be found optimal in (Liu et al., 2023). In Section 3 we also utilize deterministically generated pairing, but suggest a way to avoid this drift.

## 2.1 Optimal Transport Pairing Asymptotic Analysis

The non-negligible association between *every* pair of samples $x_0 \sim p_0$ and $x_1 \sim q$ when marginalizing the denoising losses over an independent distribution $p_0(x_0)q(x_1)$, is a common thread shared by all the key models mentioned above, undermining their sampling efficiency. By linking the flow's transport optimality to the straightness of the trajectories, both (Pooladian et al., 2023) and (Tong et al., 2024) derive their pairing between $p_0$ and $q$ from an Optimal Transport (OT) objective. Due to the cubic complexity of this problem (Flamary et al., 2021) (or a quadratic approximation (Altschuler et al., 2017)) the pairing, or plan $j_i$, is computed within batches of samples $\{x_0^i\}_{i=1}^n \sim p_0$ and $\{x_1^i\}_{i=1}^n \sim q$ of moderate sizes ($n = 50/256$ in (Pooladian et al., 2023)), and Eq. 2 is minimized over these permuted pairs. As shown in Table 1 this approach, called Batch OT CFM (BOT-CFM), results in flows that are less curved than those produced by (Lipman et al., 2023). Indeed, a 30% to 60% reduction in sampling cost is reported in (Pooladian et al., 2023).

In a well-known manifestation of the curse-of-dimensionality, the ratio between the farthest and closest points converges to a constant as the space dimension increases (Beyer et al., 1999). This suggests that

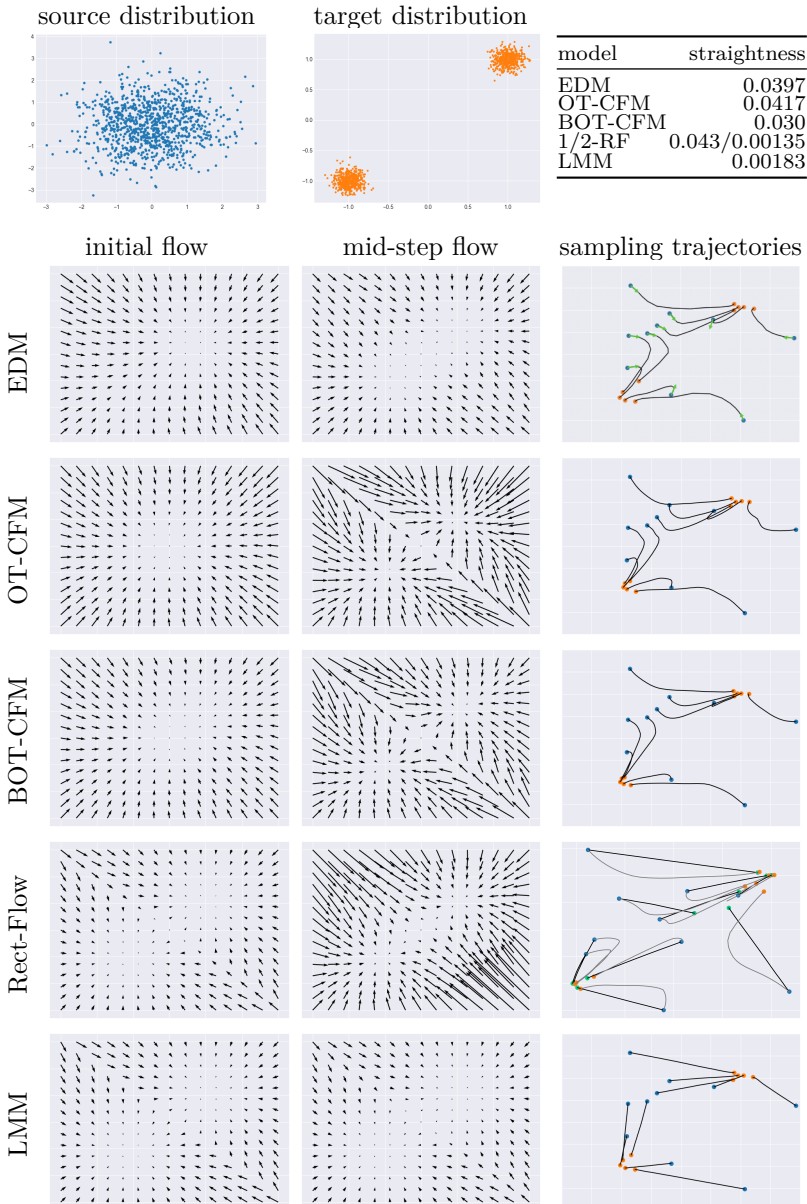

Table 1: Flow fields and sampling trajectories of different models. Top row shows the source and target distributions along their first two dimensions. The source (blue dots) is a normal distribution in 128 space dimension. The target (orange dots) is a mixture of two Gaussians located in $(-1, -1, \vec{0})$ and $(1, 1, \vec{0})$ with STDs of $(0, 1, 0.1, \vec{1})$, i.e., two separate Gaussians in the first two dimensions shown in the figure, and a normal Gaussian in the remaining 126 dimensions. The following three rows show results of the optimized DSM approach of EDM (Karras et al., 2022), the OT-CFM (Lipman et al., 2023) and its mini-batch optimized BOT-CFM (Pooladian et al., 2023) which all appear to produce curved trajectories, with an improvement observed in the BOT-CFM when pairing batches of size 256. Nearly identical results are obtained using a batch size of 128, differing in straightness by only 0.0017. The trajectories of the 1-Rect-Flow (Liu et al., 2023), shown in gray in the next row, also appear curved. The 2-Rect-Flow trajectories (black) are considerably straighter than any of the above. However, a discrepancy between these two iterations can be seen in their (target) endpoints (orange and cyan dots). This may indicate a drift from the original distribution $q$. Our LMM produces straight curves and flow Fields which are close to being constant in time. Note that excluding 2-Rect-Flow and LMM, the initial flow fields of all the methods show a clear basin of attraction at $(0, 0, \vec{0})$ responsible for an undesired drift at the beginning of the trajectories towards this point. This effect is illustrated in the EDM, where the green arrows represent the tangent vectors to the curves at their initial step. Top-right table reports the average trajectory straightness score, $\int_0^1 \|\dot{x}(t) - (x_1 - x_0)\| dt$, of each method where both 2-Rect-Flow and LMM standout.

large batches may be required for finding meaningful plans $\pi_{ij}$, as shown empirically in Kim et al. (2023). In order to analyze this relation, we consider the problem of transporting a high-dimensional unit Gaussian distribution to itself. While the solution to this problem is trivial—an identity mapping resulting from a null flow field minimizing the OT objective—we formally show the prohibitive computational requirements involved in attaining it using OT-based pairing.

The solution for the BOT-CFM objective at $t = 0$ is given by

$$N_\theta(x_0, 0) = \mathbb{E}_{p^B(x_1^*|x_0)}\big[x_1^* - x_0\big], \tag{5}$$

where $p^B(x_1^*, x_0)$ is the joint distribution induced by finding the optimal pairing between source $x_0^i$ and target $x_1^{ji}$ (denoted by $x_1^*$) within batches $B$ of size $n$. We note that the particular time $t = 0$ is of particular interest following the finding discussed in the previous section, where the effect of the attraction basins peaks. While ideally this solution (velocity) should vanish, the following proposition presents a fairly stringent lower bound. The expected maximal similarity between two Gaussian samples $x^i, x^j \sim \mathcal{N}(0, I)$, and $x^i, x^j \in \mathbb{R}^d$, found within a mini-batch $B$ of size $n$ is bounded by

$$\mathbb{E}_{p^B(x_1^*|x_0)}\big[x_1^* - x_0\big] \geq \|x_0\|^2 - 2\mathbb{E}_{p^B(x_1^*|x_0)}\big[\langle x_1^*, x_0\rangle\big] \approx 1 - 2\sqrt{\frac{2 \log n}{d}}. \tag{6}$$

Essentially, this lower bound results from the fact that the dot-product between two random vectors, $x_0^i$ and $x_1^j$, concentrates around zero as $d$ increases, and the batch-size $n$ offers a weak competition in increasing the search repertoire. The proof of both propositions is given in Appendix B.

To conclude, the exponential dependency found between the batch size $n$ and the space dimension $d$, which tends to be fairly large in practical settings, undermines the prospect of achieving additional substantial improvement beyond the one reported in (Pooladian et al., 2023) by increasing the batch size and relying solely on the BOT strategy. Table 1 shows that while at $d = 128$ the BOT-CFM shows a moderate reduction in the average trajectory straightness compared to the CFM when using batch sizes of $n = 128$, the use of $n = 256$ offers a negligible improvement.

## 3 Lines Matching Model

Utilizing pre-computed deterministic ODE-based flows, e.g. (Song et al., 2021a; Song & Ermon, 2019; Karras et al., 2022), offers an alternative to the use of random or OT-based pairing. In this empirical experimentation section, we would like to answer the following questions: Q1. what is the impact of the false basins over the sampling efficiency, Q2. should we simply distill the endpoints of the flow with a single feedforward model, or should we model the flow and operate iteratively (similarly to the teacher model's application), and Q3. can we avoid distilling the inefficient, curved trajectories in the teacher model, which current distillation methods inherit?

We answer these questions empirically, starting with Q3 by deriving a flow model that ignores the teacher flow paths, which existing models distill (Salimans & Ho, 2022; Song et al., 2023; Kim et al., 2024). By contrast, we use only the pairing they induce between the source and target distributions to construct *globally straight* flow lines to connect the distributions. Specifically, our Lines Matching Model (LMM) $N_\theta$ is derived in accordance with the VE probability flow ODE formulation used in (Karras et al., 2022), and it is trained to minimize

$$\mathcal{L}_{\text{lines}} = \mathbb{E}_{\sigma, \delta(x_1, \psi^*(x_0)), p_0(x_0)}\Big[\big\|N_\theta(x_1 + \sigma x_0, \sigma) - x_1\big\|_{\mathcal{P}}\Big], \tag{7}$$

The pairing function $\psi^*$ is inferred from a *deterministic* ODE-based sampling procedure $x_1 = N_{\text{Sampler}}^*(x_0)$ given a pre-trained denoising network $N^*$. In our implementation we use the DSM described in (Karras et al., 2022), commonly known as Elucidating Diffusion Models (EDM), along with its multi-stepped deterministic sampling procedure $N_{\text{Sampler}}^*$ that gradually reduces the noise level $\sigma$ in $x_1 + \sigma_{\max} x_0 \approx \sigma_{\max} x_0$, down to a negligible level where $x_1 + \sigma_{\min} x_0 \approx x_1$ (details in the Appendix E). Let us discuss the desirable properties of the LMM, further develop it, and address the questions raised above.

**Unambiguous Pairing.** As elaborated in the previous section, training that ties every $x_0 \sim p_0$ with every $x_1 \sim q$ by conditioning the models on $x_1$ and marginalizing over this variable leads to unwanted detours in the flow map trajectories. The deterministic pairing we use, $x_1 = N^*_{\text{Sampler}}(x_0)$ for every $x_0 \sim p_0$, corresponds to example pairs $x_0, x_1$ that sample an implicit change-of-variable function $x_1 = \psi^*(x_0)$ induced by $N^*_{\text{Sampler}}(x_0)$ and $N^*$. Thus, given a state-of-the-art $N^*$ generating samples of superior quality, the mapped distribution can be considered as a good approximation, $p_{N^*_{\text{Sampler}}} \approx q$, in this respect. Consequently, Eq. 7 regresses $N_\theta$ under a well-defined and *unambiguous pairing* between the source and target distributions *regardless* of the severity of the noise level $\sigma$.

By training this vanilla LMM we answer Q1 conclusively based on the results reported in Tables 3, 4 and 5, where it achieves FID 3.3 using NFE=3 on CIFAR-10 compared to DDPM that achieves 3.17 using NFE=1000, and OT-CFM that achieves FID 6.35 using NFE=142. Similarly, the LMM achieves FID 4.7 using NEF=3 over ImageNet 64×64while DDPM obtains FID 11 using NE=250, and OT-CFM an FID of 14.45 using NFE=138. We note that the teacher model EDM achieves lower FID scores for reasons that we discuss below. The LMM however improves on these scores by further augmenting its training as we explain below.

**Globally Straight Trajectories.** While the data distribution may be highly complicated and differ substantially from the Gaussian source, the parameterization $N^*_{\text{Sampler}}(x_0)$ provides a sample-wise paring between the two distributions. Thus, we can consider the lines that connect these endpoints. Specifically, assuming $N_\theta$ is sufficiently expressive, and satisfies Eq. 7 sufficiently well, then the lines $x_1 + \sigma_t x_0$ corresponds to its iso-contours. Thus, $N_\theta$ encodes globally straight probability flow lines connecting the source and target distributions. As noted above, while certain constructions of conditional flow maps may consist of globally straight flows Lipman et al. (2023), this property is lost once they are marginalized over $x_1$.

The validity of such linear fitting raises two important concerns. First, it is not clear that the provided pairing admits linear interpolation, more specifically, the availability of training pairs $x_0, N^*_{\text{Sampler}}(x_0)$ and $x'_0, N^*_{\text{Sampler}}(x'_0)$ whose connecting segments intersect in both time and space, i.e., $(1-t)x_0 + tN^*_{\text{Sampler}}(x_0) = (1-t)x'_0 + tN^*_{\text{Sampler}}(x'_0)$. In this case the regression in Eq. 7 is likely to result in a compromised intermediate solution, hinting that the preferable answer to Q2 above is distilling the end-to-end pairing relation without modeling any flow field.

However, the second concern relates to the fact that the input pairing is produced by a recursive application of the teacher model, and distilling this combined operator by a single feed forward model will be undermined by a mismatch in the functional space spanned by these different operators. Analogously, one could wonder why the deterministic pairing admits a linear interpolation, the lines matched by the LMM, with no intersection ambiguities. These concerns suggests that the answer to Q2 above is a construction of a flow field, that will allow multiple executions of the models to better mimic the way the input data was produced.

The experiments reported in Section 4 and Appendix C show that the LMM achieves excellent FID scores over a number of benchmark datasets, using a single NFE. This indicates that while these concerns are valid at a theoretical level, a single feed-forward execution of a model parameterizing a straight path has a good ability in practice to approximate the outcomes of a recursive application expressing curved trajectories. At the same time, all the experiment reported show an improved performance when using more NFEs, suggesting that this functional approximation can be made more accurate by deriving a flow model despite the first concern.

In either case, the sample quality achieved by the LMM falls below its teacher models, arguably due to both concerns discussed here. This limitation is overcome by the following modifications.

**Domain-Specific Loss.** Another benefit of matching $x_1$ in Eq. 7, rather than flow vectors, such as $x_1 - x_0$ as done in (Lipman et al., 2023; Liu et al., 2023), allows us to exploit the fact that $N_\theta$ matches native signals, and hence domain-specific metrics can be employed. In particular, this allows us to use the perceptual loss in (Johnson et al., 2016) to define $\|\cdot\|_{\mathcal{P}}$, when $q$ corresponds to a distribution of images. Indeed in the Appendix C we compare the use of this metric to $L_2$ loss, and show a substantial improvement in sampling fidelity lowering the FID over the CIFAR-10 dataset from 5.125 to 3.124 (NFE=1), and from 4.289 to 2.796 (NFE=2).

**Adversarial Loss.** Eq. 7 trains $N_\theta$ to replicate samples $x_1$ generated by $N^*_{\text{Sampler}}(x_0)$, rather than being trained directly on authentic (input) samples from $q$. This sets a limit on the quality at which $N_\theta$ approximates $q$—one which is bounded by the quality of the mediator network $N^*$ and its sampling procedure, $N^*_{\text{Sampler}}$. Training $N_\theta$ to produce signals in their original domain, e.g., clean images, in Eq. 7, offers yet another advantage; we can follow the strategy of (Kim et al., 2024) and bootstrap $N_\theta$ to the original training data using an adversarial loss. Specifically, we train a discriminator network $D$ to discriminate between authentic training samples $x_1 \sim q$ and ones produced by $N_\theta$, by

$$
\begin{aligned}
\mathcal{L}_{\text{disc}} = \mathbb{E}_{q(x_1)} \Big[ \log \big( D(x_1) \big) \Big] + \\
\mathbb{E}_{\sigma, \delta(x_1, \psi^*(x_0)), p_0(x_0)} \Big[ \log \Big( 1 - D\big( N_\theta (x_1 + \sigma x_0, \sigma) \big) \Big) \Big],
\end{aligned}
\tag{8}
$$

where we use the architecture in (Sauer et al., 2022) and the adaptive weighing $\lambda_{\text{adapt}}$ in (Esser et al., 2021) that Kim et al. use. We finally train $N_\theta$ to minimizing $\lambda_{\text{lines}} \mathcal{L}_{\text{lines}} + \lambda_{\text{adapt}} \mathcal{L}_{\text{disc}}$. We provide all the implementation details in Appendix E.

As we show in Appendix C, training the LMM without $\mathcal{L}_{\text{disc}}$ achieves fairly satisfying FID scores, namely 3.12 (NFE=1) and 2.79 (NEF=2) on CIFAR-10. By incorporating the latter these scores further improve to 1.67 (NFE=1) and 1.39 (NFE=2). This *surpasses* the quality of samples generated by $N^*_{\text{Sampler}}(x_0)$ which uses more sampling steps (NFE=35) and achieves FID of 1.79.

Finally, we note that 2-Rect-Flow Liu et al. (2023) also trains on deterministic pairs produced by its previous step. However, it lacks the ability to use a domain-specific reconstruction loss as it matches the flow vectors, and similarly, it does not admit a bootstrapping to the original data, which leads to the distributional drift, shown in Table 1. We attribute these differences to the significantly better scores the LMM achieves in Section 4. The integration of perceptual loss into Rect-Flow indeed offers the expected improvement, as also shown in Lee et al. (2024).

**Sampling-Optimized Training (SOT).** An additional improvement to this scheme is motivated by the fact that the LMM achieves high-quality samples already at NFE $\leq 3$, as reported in the Appendix C. Thus we explored the option of further improving its performance by restricting the training of $N_\theta$ to the specific steps (noise levels $\sigma$) used at sampling time. In Appendix C we report the further quality increase obtained by this training strategy.

## 4 Evaluation and Comparison

We trained the LMM on three benchmark datasets, CIFAR-10, ImageNet 64×64, and AFHQ 64×64, which are commonly used for evaluating generative models. We used the same network architecture and hyperparameters as existing models, with all the implementation details provided in Appendix E. *We submitted the code used to produce the results reported here, and will make it available online.*

**Quantitative Comparison.** Table 2 provides a comprehensive comparison of the CIFAR-10 reproduction quality achieved by different models. The comparison shows that diffusion-based models achieve lower FID scores, albeit at an increased sampling cost compared to GANs. Flow-matching models demonstrate an ability to reduce the NFEs, alongside a range of distillation techniques that operate effectively with very low NFEs—one or two sampling steps. With this notable difference, the LMM validates the sufficiency of deterministic pairing to avoid the inefficiencies stemming from the random denoising loss.

Among these, the Consistency Trajectory Model (CTM) (Kim et al., 2024), achieves excellent FID scores of 1.73 (NFE=1) and 1.63 (NFE=2) on conditional CIFAR-10. Our LMM surpasses these scores and sets new state-of-the-art scores of 1.57 and 1.39 respectively. We note that both methods benefit from the use of an adversarial loss, but as reported in Appendix E, the LMM's performance remains better also without this loss. We attribute this to the fact that the LMM produces favorable line flow trajectories, rather than relying on the curved EDM trajectories that the CTM distills.

The Rect-Flow in (Liu et al., 2023) achieves an impressive FID score of 4.85 in its second iteration, where it produces significantly straighter trajectories (seen in Table 1). This second iteration achieves the best trade-off between straightness and drift that this scheme accumulates. Improved scores are obtained by integrating LPIPS loss in Lee et al. (2024). We also report CLIP-FID scores for several methods, which show a consistent

**CIFAR-10**

| Model | NFE | uncond. | | conditioned |
|---|---|---|---|---|
| | | FID | IS | FID |
| **GAN** | | | | |
| BigGAN Brock et al. (2019) | 1 | 14.70 | 9.22 | – |
| StyleGAN2-ADA Karras et al. (2020) | 1 | 2.92 | 9.83 | 2.42 |
| StyleGAN-D2D Kang et al. (2024) | 1 | – | – | 2.26 |
| StyleGAN-XL Sauer et al. (2022) | 1 | – | – | 1.85 |
| **Diffusion / Score Matching** | | | | |
| DDPM Ho et al. (2020) | 1000 | 3.17 | 9.46 | – |
| DDIM Song et al. (2021a) | 100 | 4.16 | – | – |
| Score SDE Song et al. (2021b) | 2000 | 2.20 | 9.89 | – |
| EDM Karras et al. (2022) | 35 | 1.97 | 9.84 | 1.79 |
| ↪ CLIP-FID | 35 | 0.55 | – | 0.50 |
| **Distillation / Direct Gen.** | | | | |
| KD Luhman & Luhman (2021) | 1 | 9.36 | 8.36 | – |
| PD Salimans & Ho (2022) | 1 | 9.12 | – | – |
| CT Song et al. (2023) | 1 | 8.70 | 8.49 | – |
| CD Song et al. (2023) | 1 | 3.55 | 9.48 | – |
| ↪ CLIP-FID | 1 | 1.26 | – | – |
| CD+GAN Lu et al. (2023) | 1 | 2.65 | – | – |
| iCT Song & Dhariwal (2024) | 1 | 2.83 | 9.54 | – |
| iCT-deep Song & Dhariwal (2024) | 1 | 2.51 | 9.76 | – |
| CTM Kim et al. (2024) | 1 | 1.98 | – | 1.73 |
| DMD Yin et al. (2024) | 1 | 3.77 | – | 2.66 |
| SiD ($\alpha = 1$) Zhou et al. (2024) | 1 | 2.02 | 10.02 | 1.93 |
| SiD ($\alpha = 1.2$) Zhou et al. (2024) | 1 | 1.92 | 9.98 | 1.71 |
| ↪ CLIP-FID | 1 | 0.65 | – | – |
| PD Salimans & Ho (2022) | 2 | 4.51 | – | – |
| CT Song et al. (2023) | 2 | 5.83 | 8.85 | – |
| CD Song et al. (2023) | 2 | 2.93 | 9.75 | – |
| iCT Song & Dhariwal (2024) | 2 | 2.46 | 9.80 | – |
| iCT-deep Song & Dhariwal (2024) | 2 | 2.24 | 9.89 | – |
| CTM Kim et al. (2024) | 2 | 1.87 | – | 1.63 |
| **Flow Matching** | | | | |
| OT-CFM Lipman et al. (2023) | 142 | 6.35 | – | – |
| 1-Rect-Flow (distill) Liu et al. (2023) | 1 | 6.18 | 9.08 | – |
| 2-Rect-Flow (distill) Liu et al. (2023) | 1 | 4.85 | 9.01 | – |
| 3-Rect-Flow (distill) Liu et al. (2023) | 1 | 5.21 | 8.79 | – |
| Simple-ReFlow Kim et al. (2025) | 9 | 2.23 | – | – |
| 2-Rect-Flow++ Lee et al. (2024) | 1 | 3.07 | – | – |
| 2-Rect-Flow++ Lee et al. (2024) | 2 | 2.40 | – | – |
| 1-Rect-Flow Liu et al. (2023) | 127 | 2.58 | 9.60 | – |
| 2-Rect-Flow Liu et al. (2023) | 110 | 3.36 | 9.24 | – |
| 2-Rect-Flow Liu et al. (2023) | 104 | 3.96 | 9.01 | – |
| LMM (NFE=1) | 1 | **1.90** | **10.16** | **1.57** |
| ↪ CLIP-FID | 1 | 0.63 | – | 0.43 |
| LMM (NFE=2) | 2 | **1.55** | **10.20** | **1.39** |
| ↪ CLIP-FID | 2 | 0.52 | – | 0.38 |

**ImageNet64**

| Model | NFE | conditional | |
|---|---|---|---|
| | | FID | IS |
| **GANs** | | | |
| BigGAN-deep Brock et al. (2019) | 1 | 4.06 | – |
| StyleGAN-XL Sauer et al. (2022) | 1 | 1.51 | 82.35 |
| **Diffusion / Score Matching** | | | |
| RIN Jabri et al. (2023) | 1000 | 1.23 | – |
| EDM Karras et al. (2022) | 511 | 1.36 | – |
| DDPM Ho et al. (2020) | 250 | 11 | – |
| EDM Karras et al. (2022) | 79 | 2.23 | 48.88 |
| **Distillation / Direct Gen.** | | | |
| PD Salimans & Ho (2022) | 1 | 15.39 | – |
| BOOT Gu et al. (2023) | 1 | 16.30 | – |
| CT Song et al. (2023) | 1 | 13.0 | – |
| CD Song et al. (2023) | 1 | 6.20 | 40.08 |
| iCT Song & Dhariwal (2024) | 1 | 4.02 | – |
| iCT-deep Song & Dhariwal (2024) | 1 | 3.25 | – |
| CTM Kim et al. (2024) | 1 | 1.92 | 70.38 |
| DMD Yin et al. (2024) | 1 | 2.62 | – |
| SiD ($\alpha = 1$) Zhou et al. (2024) | 1 | 2.02 | – |
| SiD ($\alpha = 1.2$) Zhou et al. (2024) | 1 | 1.52 | – |
| PD Salimans & Ho (2022) | 2 | 8.95 | – |
| CT Song et al. (2023) | 2 | 11.1 | – |
| CD Song et al. (2023) | 2 | 4.70 | – |
| iCT Song & Dhariwal (2024) | 2 | 3.20 | – |
| iCT-deep Song & Dhariwal (2024) | 2 | 2.77 | – |
| CTM Kim et al. (2024) | 2 | 1.73 | 64.29 |
| **Flow Matching** | | | |
| OT-CFM Lipman et al. (2023) | 138 | 14.45 | – |
| BOT-CFM Pooladian et al. (2023) | 132 | 11.82 | – |
| 2-Rect-Flow++ Lee et al. (2024) | 1 | 4.31 | – |
| 2-Rect-Flow++ Lee et al. (2024) | 2 | 3.64 | – |
| Simple-ReFlow Kim et al. (2025) | 9 | 1.74 | – |
| LMM | 1 | **1.47** | 59.86 |
| LMM | 2 | **1.17** | 61.18 |

**AFHQ64**

| Model | NFE | FID |
|---|---|---|
| EDM Karras et al. (2022) | 79 | 1.96 |
| SiD ($\alpha = 1.2$) Zhou et al. (2024) | 1 | 1.71 |
| SiD ($\alpha = 1$) Zhou et al. (2024) | 1 | 1.63 |
| Simple-ReFlow Kim et al. (2025) | 9 | 1.91 |
| LMM | 1 | 2.68 |
| LMM | 2 | 1.54 |

Table 2: One-step generative model performance on CIFAR-10, ImageNet-64, and AFHQ64 (all measured with FID/IS as shown).

trend with FID, addressing the concern that the latter might favor VGG-based training solutions, since both metrics were trained on ImageNet. As explained in Section 3, the LMM offers additional improvements to this model, namely, the use of adversarial loss to avoid drifting away from the target distribution, as well as the sampling optimized training (SOT) strategy that focuses the training on the actual sampling steps used in practice.

Finally, let us note that by contrast to end-to-end samples distillation methods Luhman & Luhman (2021); Yin et al. (2024); Zhou et al. (2024), by modeling a probability flow the LMM allows for improving sampling quality by applying additional sampling steps. Results using NFE > 2 are reported in Appendix C.

Table 2 shows the results obtained on a larger dataset, ImageNet 64×64. Here too, the LMM demonstrates state-of-the-art performance, with a notable improvement at NFE=2, where it reaches an FID of 1.17. The SiD (Zhou et al., 2024) trains a single-step generator to agree with a pre-trained EDM, achieving an impressive score of 1.52. Unlike the LMM, SiD does not rely on the EDM to generate training examples; instead, it uses it to define the generator's loss while simultaneously training an additional score-matching network. This approach poses significantly higher GPU memory requirements and operations during training.

Finally, Table 2 also reports the results on the AFHQ 64×64 dataset, where the LMM shows lower FID scores using significantly fewer NFEs compared to the EDM despite the fact that the latter is used to produce the initial correspondence between $p_0$ and $q$. This is also the case in CIFAR-10 and ImageNet 64×64. While achieving a state-of-the-art FID of 1.54 at NFE=2, the SiD achieves a better score using a single step. We note that unlike the CIFAR-10 and ImageNet 64×64 cases, the discriminator architecture and hyper-parameters we used were not we used were not tailored to this dataset in previous work (e.g., StyleGAN-XL (Sauer et al., 2022)). This affected the expected improvements from the SOT strategy, as discussed in Appendix C, and we therefore believe the LMM has greater potential on this dataset.

In terms of Inception Score (IS), the LMM achieves state-of-the-art results, scoring above 10 for both NFEs on CIFAR-10, as shown in the table. On ImageNet 64×64, the LMM improves upon its teacher model (EDM), although StyleGAN-XL attains the highest score. Among diffusion-based models, the LMM receives an IS of 61.18 using 2 NFEs, which is closely competitive with CTM which scores 64.29. A visual comparison and ablation of LMM samples can be found in Appendix C and D.

## 5    Conclusions

We analyzed the impact of random pairing in the denoising loss and identified basins of attraction that curve the probability flow paths, and thus increase the sampling costs in key denoising models. We also formally showed that OT-based pairing suffers from unfavorable batch-size scaling with signal dimension, limiting its practicality in high-dimensional settings.

Motivated by these results, we explored strategies for modeling pre-computed deterministic ODE-based flows and showed the importance of selectively utilizing this data to avoid inheriting inefficiencies. This led to the Lines Matching Model (LMM), which matches straight lines between corresponding source and target flow points. The LMM's performance is further improved by bootstrapping it to the target distribution and avoiding distributional shift seen in existing models, as well as by optimizing the training to the particular sampling scheme used.

Our work leaves one important goal unaddressed that we intend to address in future work: avoid the reliance on a pre-trained model, and establish or employ "cheaper" pairing in an ab initio manner.

## Broader Impact Statement

Given their increasing prevalence, improving the efficiency of generative AI models is likely to result in a significant reduction in computational costs and energy usage. However, we are fully aware of the risks associated with these models and wish to express our strong opposition to any unethical use.

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

## Appendix

## A  Regression at Low Signal-to-Noise Ratios

In this section we provide the derivation of section 2, and its consequence to DDPM, NCSN and CFM. Both DDPM (Sohl-Dickstein et al., 2015; Ho et al., 2020) and DSM, e.g., NCSN (Song & Ermon, 2019) and EDM (Karras et al., 2022), start their sampling process from an easy-to-sample source distribution $x^N \sim p_0$, typically a Gaussian. Hence their denoising networks $N_\theta$ are trained to operate on these distributions. At $t = T$, Eq. 1 becomes

$$\mathrm{argmin}_\theta \mathbb{E}_{q(x_1),p(x|x_1,T)}\Big[\|N_\theta(x,s_T) - \nabla_x \log p(x|x_1,T)\|^2\Big]. \tag{9}$$

In the case of NCSN, $s_T = \sigma_T$ and $p(x|x_1,\sigma_T) = \mathcal{N}(x_1,\sigma_T^2 I)$ where $\sigma_T^2 >> \mathbb{V}[x_1]$, i.e., a very low SNR, allowing to approximate this distribution by a pure noise at sampling time, specifically $p_0 = \mathcal{N}(0,\sigma_T^2 I)$.

Noting that $\nabla_x \log p(x|x_1,T) = (x_1 - x)/\sigma_T^2$, Eq. 9 becomes (at $t = T$),

$$\mathrm{argmin}_\theta \mathbb{E}_{q(x_1),x_0 \sim p_0}\Big[\|N_\theta(x_0,\sigma_T) - (x_1 - x_0)/\sigma_T^2\|^2\Big], \tag{10}$$

which corresponds to a regression problem solved by

$$N_\theta(x_0,\sigma_T) = \mathbb{E}_{q(x_1)}\Big[(x_1 - x_0)/\sigma_T^2\Big]. \tag{11}$$

In principle one may be concerned whether this optimal solution is attainable by $N_\theta$ and how the weighting $p_0(x_0)$ affect the solution. As we show next, the solution in this context is highly degenerate and corresponds to an affine function which is expected to lie within the span of any elementary architecture of $N_\theta$.

Specifically, Eq. 10 trains the network $N_\theta$ to map every $x_0 \sim p_0$ to every $x_1 \sim q$. Regressing under such indeterminacy is poised, according to Eq. 11, to result in the degenerate averaged prediction $N_\theta(x,\sigma_T) = (\mu_1 - x)\sigma_T^{-2}$. This topic is discussed in (Kendall & Stuart, 1973; Clarke & Gorder, 2013). Finally, we note that at sampling stage the factor $\sigma_T^{-2}$ is typically canceled by using time steps proportional to $\sigma_T^2$, see for example (Song & Ermon, 2019) and (Karras et al., 2022). Thus, the sampling trajectories are drawn towards $\mu_1$, up to some implementation-dependent speed factors, during their first steps. This effect is highlighted by the green arrows in Table 1.

Analogously, the DDPM noise scheduling is set such that $\alpha_T$ is small, e.g., $\alpha_T = 6 \times 10^{-3}$ in (Ho et al., 2020) and $\alpha_T = 5 \times 10^{-5}$ in (Nichol & Dhariwal, 2021). Therefore $p(x|x_1,T) = \mathcal{N}(\sqrt{\alpha_T}x_1,(1-\alpha_T)I) \approx \mathcal{N}(0,I)$

which, here as well, can be replaced with the source distribution $p_0$ during sampling. In this case, Eq. 9 becomes (again, at $t = T$),

$$\text{argmin}_\theta \mathbb{E}_{q(x_1), x_0 \sim p_0} \Big[ \| N_\theta(x_0, T) - (\sqrt{\alpha_T} x_1 - x_0)/(1 - \alpha_T) \|^2 \Big], \tag{12}$$

resulting in $N_\theta(x, T) = (\sqrt{\alpha_T} \mu_1 - x)/(1 - \alpha_T)$. Recall that the DDPM noising process $p(x|x_1, T) = \mathcal{N}(\sqrt{\alpha_t} x_1, (1 - \alpha_t)I)$ gradually replaces every data sample $x_1$ with a normal Gaussian by shifting the mean from $x_1$ towards $\mu_0$ (chosen to be 0 for convenience) and by increasing the variance from $(1 - \alpha_1) \approx 0$ to $\mathbb{V}[x_0] = 1$. Thus, this term can be more generally interpreted as $N_\theta(x, T) = (\sqrt{\alpha_T} \mu_1 + \sqrt{(1 - \alpha_T)} \mu_0 - x)/(1 - \alpha_T) \approx \mu_0 - x$, since $\alpha_T << 1$.

Thus, at the $T$-th step of the DDPM sampling step, the denoiser collapses to the mean of the source distribution. Consequently, its flow trajectories gravitate toward $\mu_0 = 0$ during their earlier steps. This affects only the magnitude of initial (full noise) states, and the sample's shape evolves only in later steps, thus the DDPM sampling process is often described as stagnant during its early stages (e.g., in Figure 6 in (Lipman et al., 2023)).

Finally, the flow models in (Lipman et al., 2023), and (Liu et al., 2023) at 1-Rect-Flow, regress arbitrary samples from $p_0$ to the data points $x_1$ at time $t = 0$, where its training loss, Eq. 13, becomes

$$\text{argmin}_\theta \mathbb{E}_{q(x_1), p(x_0)} \Big[ \| N_\theta(x_0, 0) - (x_1 - x_0) \|^2 \Big], \tag{13}$$

Similarly to Eq. 10, also Eq. 13 regresses points $x_0$ with direction towards arbitrary data points, $x_1 - x_0$. This leads again to a degenerate solution where $N_\theta(x_0, 0) = \mu_1 - x_0$, which similarly to the score-matching approach, biases the sampling trajectories towards $\mu_1$ at their earlier stages. This effect is also observed in Table 1.

## B  Batch Optimal Transport - Batch Size Analysis

We assess here the asymptotic dependence in the Batch OT CFM (BOT-CFM) methods described in (Pooladian et al., 2023) and (Tong et al., 2024) over the batch size $n$ as a function of space dimension $d$, and prove Proposition 2.1. In these works, following the notations of the former, the independent distribution $p_0(x_0)q(x_1)$ in Eq 2 is replaced by a joint distribution $q(x_0, x_1)$ induced by a batch-optimized coupling, $\{x_0^i\}_{i=1}^n \sim p_0$ and $\{x_1^{j_i}\}_{i=1}^n \sim q$, where the permutation $j_i$ optimizes the transport cost $\|x_0^i - x_1^{j_i}\|^2$ within each batch. Combining this with the OT conditional flow map $\psi_{x_1}(x, t)$ in (Lipman et al., 2023), the BOT-CFM training loss is given by

$$\text{argmin}_\theta \mathbb{E}_{t, \{x_0^i\}_{i=1}^n \sim p_0, \{x_1^i\}_{i=1}^n \sim q} \Big[ \sum_{i=1}^n \| N_\theta((1 - t)x_0^i + tx_1^{j_i}, t) - (x_1^{j_i} - x_0^i) \|^2 \Big], \tag{14}$$

To simplify the analysis we consider a fairly naive problem of finding a mapping from a normal Gaussian in $\mathbb{R}^d$ to itself, where the optimal solution is given by the identity mapping. In the context of matching the velocity field, as done in (Pooladian et al., 2023; Tong et al., 2024), the optimal field is given by $N_\theta(x, t) = 0$. As shown in Appendix A, in case of independent distribution $p_0(x_0)q(x_1)$ (the solution of Eq. 13)) the resulting vector field at $t = 0$ is $N_\theta(x_0, 0) = \mu_1 - x_0 = -x_0 \neq 0$ which is clearly far from the optimum.

In the BOT-CFM (at $t = 0$) closer and closer $x_1^{j_i}$ will be found to each $x_0^i$ as the batch size increases, and hence by training $N_\theta(x_0^i, 0)$ to match $x_1^{j_i} - x_0^i$, in Eq. 14, a reduced velocity vector is expected. The question of how fast this decrease takes place as a function of $d$ is critical, as only moderately sized batches can be used in practice.

We address this question at $t = 0$, where Eq. 14 simplifies to a simple regression problem over $x_0$,

$$\text{argmin}_\theta \mathbb{E}_{\{x_0^i\}_{j=1}^n \sim p_0, \{x_1^i\}_{i=1}^n \sim q} \Big[ \sum_{i=1}^n \| N_\theta(x_0^i, 0) - (x_1^{j_i} - x_0^i) \|^2 \Big], \tag{15}$$

which is solved by,

$$N_\theta(x_0, 0) = \mathbb{E}_{p^{B_1^n}(x_1^*|x_0)}[x_1^* - x_0], \tag{16}$$

where $p^{B_1^n}(x_0, x_1^*)$ is the joint distribution induced by finding the optimal pairing between source $x_0^i$ and target $x_1^{j_i}$ within each batch $B_1^n$ of size $n$.

The case $n=1$ (equivalent to random pairing), we get $p^{B_1^1}(x_1^*|x_0) = p_0(x_0)q(x_1)$ which was discussed above and results in a velocity $N_\theta(x_0, 0) = -x_0$ attracting sampling trajectories towards $\mu_1 = 0$ at $t=0$, instead of remaining stationary, thus producing the unnecessarily curved trajectories. As $n$ increases, however, the chances to regress $x_0$ to closer $x_1^*$ increases and thus a shift in $\mathbb{E}_{p^{B_1^n}(x_1^*|x_0)}[x_1^*]$ toward $x_0$ is expected. In order to analyze the magnitude of this shift as a function of both $n$ and $d$, let us review basic properties of random vectors in $\mathbb{R}^d$.

Let $x$ and $y$ be two independent normal scalars drawn from $\mathcal{N}(0,1)$. Their product $xy$ is a random variable with the following moments

$$\mathbb{E}[xy] = \mathbb{E}[x]\mathbb{E}[y] = 0 \tag{17}$$

and,

$$\mathbb{V}[xy] = \mathbb{E}[(xy)^2] = \mathbb{E}[x^2]\mathbb{E}[y^2] = \mathbb{V}[x]\mathbb{V}[y] = 1 < \infty, \tag{18}$$

both follow from the normality and independence of $x, y$. Let us consider now two independent normal vectors $x, y \in \mathbb{R}^d$, drawn from $\mathcal{N}(0, I)$, and their dot-product, defined by

$$\langle x, y \rangle = \frac{1}{d} \sum_{i=1}^d x^i y^i. \tag{19}$$

Being an average of independent random variables, at large space dimension $d$ the central limit theorem becomes applicable and provides us its limit distribution by,

$$\langle x, y \rangle \xrightarrow{d} \mathcal{N}(0, d^{-1}), \tag{20}$$

which is calculated from the scalar moments in Eq. 17 and Eq. 18. This implies that as the space dimension $d$ increases, this distribution gets more concentrated around 0, meaning that the vectors $x$ and $y$ are becoming less likely to be related to one another by becoming increasingly orthogonal. As we shall now show, this makes the task of finding $x_1 \in B_1^n$ close to $x_0$ within finite batches increasingly difficult as $d$ grows. This relates to a well-known phenomenon where the ratio between the farthest and closest points converges to a constant, as the space dimension increases (Beyer et al., 1999).

Indeed, by considering the magnitude of the regressed flow velocity in Eq. 16,

$$\begin{aligned}
\left\|\mathbb{E}_{p^{B_1^n}(x_1^*|x_0)}[x_1^*] - x_0\right\|^2 &= \left\|\mathbb{E}_{p^{B_1^n}(x_1^*|x_0)}[x_1^*]\right\|^2 + \|x_0\|^2 - 2\langle\mathbb{E}_{p^{B_1^n}(x_1^*|x_0)}[x_1^*], x_0\rangle \\
&\geq \|x_0\|^2 - 2\langle\mathbb{E}_{p^{B_1^n}(x_1^*|x_0)}[x_1^*], x_0\rangle = \|x_0\|^2 - 2\mathbb{E}_{p^{B_1^n}(x_1^*|x_0)}[\langle x_1^*, x_0\rangle],
\end{aligned} \tag{21}$$

we clearly see the need for increased dot-product similarity within the batches $B_1^n$ in order to reduce the magnitude of the learned target flow velocity—ideally zero in this problem. In this derivation $\left\|\mathbb{E}_{p^{B_1^n}(x_1^*|x_0)}[x_1^*]\right\|^2$ is neglected as we are in a process of deriving a lower bound for the flow velocity field, $\left\|\mathbb{E}_{p^{B_1^n}(x_1^*|x_0)}[x_1^*] - x_0\right\|^2$. We also note that the last equality follows from the linearity of the dot-product operator.

As an upper bound for $\langle\mathbb{E}_{p^{B_1^n}(x_1^*|x_0)}[x_1^*], x_0\rangle$ we assume that this similarity is computed by pairing $x_0$ with its *closest* $x_1^* \in B_1^n$ without considering trade-offs that arise when pairing a complete batch of source points $\{x_0^i\}_{j=1}^n \sim p_0$ with the batch of target points, in $B_1^n$, as done in practice in BOT-CFM, in Eq. 14.

In this scenario, $\langle x_1^*, x_0 \rangle = \max_i \langle x_1^i, x_0 \rangle$, where $\langle x_1^i, x_0 \rangle$ are independent variables and, as shown above, $\langle x_1^i, x_0 \rangle \sim \mathcal{N}(0, d^{-1})$. Using Jensen's inequality, we get that

$$\begin{aligned}
\exp\left(t\mathbb{E}_{p^{B_1^n}(x_1^*|x_0)}[\langle x_1^*, x_0\rangle]\right) &\leq \mathbb{E}_{p^{B_1^n}(x_1^*|x_0)}\left[\exp(t\langle x_1^*, x_0\rangle)\right] = \mathbb{E}_{\mathcal{N}(0, d^{-1})}\left[\max_i \exp(t\langle x_1^i, x_0\rangle)\right] \\
&\leq \sum_{i=1}^n \mathbb{E}_{\mathcal{N}(0, d^{-1})}\left[\exp(t\langle x_1^i, x_0\rangle)\right] = n\exp\left(\frac{t^2}{2d}\right),
\end{aligned} \tag{22}$$

where the last equality follows from the calculation of the moment generating function of the Gaussian distribution, $\mathcal{N}(0, d^{-1})$. Thus, by taking the logarithm of Eq. 22 and dividing by $t$ we get

$$\mathbb{E}_{p^{B_1^n}(x_1^*|x_0)}[\langle x_1^*, x_0 \rangle] \leq \log(n)/t + \frac{t}{2d}. \tag{23}$$

Finally, by setting $t = \sqrt{2d \log n}$, we get

$$\mathbb{E}_{p^{B_1^n}(x_1^*|x_0)}[\langle x_1^*, x_0 \rangle] \leq \sqrt{\frac{2 \log n}{d}}. \tag{24}$$

According to Eq. 21, in order for the flow to vanish, $\mathbb{E}_{p^{B_1^n}(x_1^*|x_0)}[\langle x_1^*, x_0 \rangle]$ should cancel $\|x_0\|^2$. For similar arguments as the ones behind Eq.19 and Eq.20, it follows that $\|x_0\|^2 = \langle x_0, x_0 \rangle \xrightarrow{d} \mathcal{N}(1, d^{-1})$. Hence, by plugging Eq. 24 into Eq. 21 and approximating $\|x_0\|^2 \approx 1$ (at high $d$), we get the following requirement for the flow to vanish

$$2\sqrt{\frac{2 \log n}{d}} \approx 1, \implies n \approx e^{d/8}. \tag{25}$$

**Conclusion.** This relation implies that in order to obtain a proper (zero) target velocity field in Eq. 21, the batch size $n$ must grow exponentially as a function of the space dimension $d$, which tends to be fairly large in practical settings. As noted in Section 2, this finding undermines the prospect of accelerating sampling by increasing the batch size and relying solely on OT pairing. Indeed, in the example shown in Table 1, a negligible difference in trajectory straightness is found between $n = 128$ and $n = 256$.

Several notes on the scope of our analysis which considered a simple problem of mapping two Gaussians and considered the affairs at $t = 0$. First, it shows that even over an arguably simple problem the effectiveness of the BOT-CFM is limited by its asymptotic. Second, as discussed at great length in Section 2 a major source of sampling inefficiency, shared by multiple key approaches, takes place at the vicinity of $t = 0$, and hence the focus of our analysis to this time should not necessarily be considered as a limitation. Finally, most of the arguments made above remain valid when real-world target data distribution $q$ is used. Namely, the limiting orthogonal distribution in Eq. 20 and hence the exponential batch size requirement for finding real-world data point $x_1^*$ sufficiently close to a random latent vector $x_0 \sim \mathcal{N}(0, d^{-1})$. Our restriction to a target Gaussian distribution is made specifically for the purpose of being able to consider the analytical results with respect to a *known* optimal flow field.

## C  Ablation Studies

We report here the results of several empirical experiments that assess the impact of different components related to LMM's training, described in Section 3, on its sampling performance and quality.

**Domain-Specific versus L2 Loss.**   Training the LMM to reproduce the end-points of the probability flow lines, i.e., noise-free images, allows us employ perceptual metrics, specifically (Johnson et al., 2016), for training. This loss is known to provide visually-preferable optimization trade-offs in various applications, see (Zhang et al., 2018). Table 3 shows that training the LMM using a VGG-based perceptual loss (VGG) achieves lower FID scores compared to that of L2 loss at all NFEs tested. The ability to use this reconstruction loss is inherent to the design of the LMM, and is not shared by all flow-based approaches, e.g., (Lipman et al., 2023; Liu et al., 2023).

**Number of Sampling Steps.**   Tables 3, 4, and 5 report the FID scores on different datasets using different NFEs and sampling steps. Specifically, we used subsets of the sampling steps from the sampling scheme in (Karras et al., 2022). While the number of steps provides some amount of ability to trade-off between quality and efficiency, it is clear from these tables that increasing the NFEs suffers from a diminishing return. This finding aligns with the explanation that the probability flow lines generated by the LMM are fairly straight, and that the sampling errors are primarily due to the accuracy of their endpoints, i.e., the quality at which the target samples $x_1 \sim q$ can be reproduced by exact integration. This further motivated us in Section 3 to focus on improving the sample reproduction, as we evaluate next.

**Adversarial Loss.** Indeed, Tables 3, 4, and 5, show that the incorporation of an adversarial loss (ADL) provides an additional significant improvement to the image quality produced by the LMM. Indeed, this addition also helped the CTM in (Kim et al., 2024) to improve their baseline, specifically, FID of 2.28 using a discriminator and 5.19 without it, using NFE=1 on CIFAR-10. We attribute the lower FID scores achieved by the LMM, in both scenarios, to the fact that it models favorable line flow trajectories, rather than the original curved EDM's trajectories, which are distilled in (Kim et al., 2024).

**Sampling-Optimized Training.** Motivated by limited improvement higher NFEs produce, in Section 3 we proposed another strategy to improve sample quality by restricting the training to the specific time steps used at the sampling stage. Tables 3 and 4 show that this training strategy also has the ability to contribute significantly despite the fact that it adds no cost. Table 5 an opposite trend which appears to be related to a saturation (over-fitting) due to two factors: (i) to limited data available in this dataset, and (ii) the SOT focuses on high noise levels, which makes it easier to discriminate between generated and real samples. We conclude that a more fine-tuned discriminator setting is needed to achieve optimal results.

| | | CIFAR-10 (conditional) | | | |
| --- | --- | --- | --- | --- | --- |
| NFE | Steps | L2 FID $\pm$ std | VGG FID $\pm$ std | VGG+ADL FID $\pm$ std | VGG+ADL+SOT FID $\pm$ std |
| 1 | 0 | $5.125 \pm 0.050$ | $3.124 \pm 0.024$ | $1.672 \pm 0.018$ | $1.575 \pm 0.016$ |
| 2 | 0, 1 | $4.289 \pm 0.032$ | $2.796 \pm 0.020$ | $1.394 \pm 0.010$ | $1.389 \pm 0.011$ |
| 3 | 0, 1, 2 | $4.019 \pm 0.026$ | $2.761 \pm 0.021$ | $1.386 \pm 0.009$ | - |
| 3 | 0, 3, 5 | $3.337 \pm 0.042$ | $2.601 \pm 0.019$ | $1.381 \pm 0.015$ | - |
| 4 | 0, 1, 3, 5 | $3.315 \pm 0.023$ | $2.625 \pm 0.025$ | $1.383 \pm 0.012$ | - |

Table 3: Selected step indices $t$ from the original EDM schedule $\sigma_t$ consisting of 18 steps for this dataset.

| | | ImageNet 64×64 (conditional) | | |
| --- | --- | --- | --- | --- |
| NFE | Steps | VGG FID $\pm$ std | VGG+ADL FID $\pm$ std | VGG+ADL+SOT FID $\pm$ std |
| 1 | 0 | $6.968 \pm 0.051$ | $1.731 \pm 0.013$ | $1.473 \pm 0.016$ |
| 2 | 0, 1 | $5.472 \pm 0.042$ | $1.318 \pm 0.013$ | $1.167 \pm 0.016$ |
| 3 | 0, 1, 2 | $5.004 \pm 0.057$ | $1.301 \pm 0.012$ | - |
| 3 | 0, 3, 5 | $4.694 \pm 0.047$ | $1.284 \pm 0.016$ | - |

Table 4: Selected step indices $t$ from the original EDM schedule $\sigma_t$ consisting of 40 steps for this dataset.

| | | AFHQ 64×64 | | |
| --- | --- | --- | --- | --- |
| NFE | Steps | VGG FID $\pm$ std | VGG+ADL FID $\pm$ std | VGG+ADL+SOT FID $\pm$ std |
| 1 | 0 | $5.458 \pm 0.053$ | $2.687 \pm 0.046$ | $2.767 \pm 0.056$ |
| 2 | 0, 1 | $4.254 \pm 0.039$ | $1.545 \pm 0.023$ | $1.776 \pm 0.022$ |
| 3 | 0, 1, 2 | $4.165 \pm 0.045$ | $1.462 \pm 0.016$ | - |
| 3 | 0, 3, 5 | $3.919 \pm 0.035$ | $1.447 \pm 0.023$ | - |

Table 5: Selected step indices $t$ from the original EDM schedule $\sigma_t$ consisting of 40 steps for this dataset.

## D   Visual Evaluation

Table 1 compares samples produced by the EDM and LMM with and without an Adversarial Loss (ADL), and using 1 or 2 sampling steps. The ADL contributes to richness and resolvedness of fine image details (e.g., the fish background, lettuce leaves, bird feathers, and man's face). The second sampling iteration (NFE=2 in the table) has a larger scale impact, improving the correctness of the objects' shape and consistency between different objects. This effect is seen in the clerk's body and face, the bird's body, the shape of the bread/cake, and the matching red shoes.

## E   Implementation Details

We implemented the LMM in PyTorch and trained it on four GeForce RTX 2080 Ti GPUs on three commonly used benchmark datasets: CIFAR-10, ImageNet 64×64, and AFHQ 64×64(aka. AFHQ-v2 64×64). We

Table 6: Network architectures and hyper-parameters used for different datasets.

| Hyper-Parameter | CIFAR-10 | AFHQ 64×64 | ImageNet 64×64 |
|---|---|---|---|
| Generator architecture | DDPM++ | DDPM++ | ADM |
| Channels | 128 | 128 | 192 |
| Channels multipliers | 2, 2, 2 | 1, 2, 2, 2 | 1, 2, 3, 4 |
| Residual blocks | 4 | 4 | 3 |
| Attention resolutions | 16 | 16 | 32, 16, 8 |
| Attention heads | 1 | 1 | 6, 9, 12 |
| Attention blocks in encoder | 4 | 4 | 9 |
| Attention blocks in decoder | 2 | 2 | 13 |
| Generator optimizer | RAdam | RAdam | RAdam |
| Discriminator optimizer | RAdam | RAdam | RAdam |
| Generator learning rate | 0.0004 | 0.0001 | 0.000008 |
| Discriminator learning rate | 0.002 | 0.002 | 0.002 |
| Generator $\beta_1, \beta_2$ | 0.9, 0.999 | 0.9, 0.999 | 0.9, 0.999 |
| Discriminator $\beta_1, \beta_2$ | 0.5, 0.9 | 0.5, 0.9 | 0.5, 0.9 |
| Batch size | 512 | 512 | 512 |
| EMA | 0.999 | 0.999 | 0.999 |
| Training images | 1M | 2M | 4M |
| Training iterations | 80k+20k w/ADL. | 80k+25k w/ADL | 80k+30k w/ADL |
| $\lambda_{\text{lines}}$ | 0.5 | 0.5 | 0.5 |

| EDM | LMM | | | |
|---|---|---|---|---|
| | wo/ADL | | w/ADL | |
| NFE 79 | NFE 1 | NFE 2 | NFE 1 | NFE 2 |

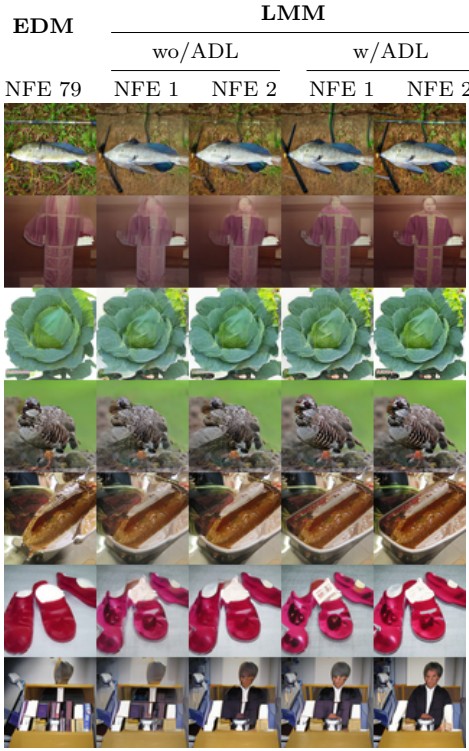

Figure 1: ImageNet 64×64 Samples Comparison.

employed the network architectures and hyper-parameters listed in Table 6, which were previously used in (Karras et al., 2022; Song et al., 2023; Kim et al., 2024; Lipman et al., 2023) over these datasets.

**Training Losses.**  As noted above, we used the VGG-based perceptual loss in (Johnson et al., 2016) as our image reconstruction loss term in Eq. 7, and resized the images to 224-by-224 pixels before evaluating it.

We use a similar adversarial loss as the one used in (Kim et al., 2024), namely, we adopted the discriminator architecture from (Sauer et al., 2022) and used its conditional version when training on labeled datasets.

We used the same feature extraction networks they use, as well as the adaptive weighing from (Esser et al., 2021), given by $\lambda_{\text{adapt}} = \|\nabla_{\theta_L}\mathcal{L}_{\text{lines}}\|/\|\nabla_{\theta_L}\mathcal{L}_{\text{disc}}\|$, where $\theta_L$ denotes the weights of the last layer of $N_\theta$. We also used their augmentation strategy, taken from (Zhao et al., 2020), we resized the images to 224-by-224 pixels before applying this loss as well.

**Denoising ODE.** As noted in Section 3, we use the EDM denoising score-matching model $N^*$ in (Karras et al., 2022) in order to produce our training pairs $x_0, N^*_{\text{Sampler}}(x_0)$. We use their deterministic sampler (second-order Heun) in order to establish a well-defined change-of-variable, $N^*_{\text{Sampler}}(x)$, between the source and target distributions. This scheme uses a source distribution $p_0 = \mathcal{N}(0, \sigma_{\text{max}})$ and noise scheduling $\sigma_t = \left(\sigma_{\text{max}}^{1/\rho} + t/(N-1)(\sigma_{\text{min}}^{1/\rho} - \sigma_{\text{max}}^{1/\rho})\right)^\rho$, where $\rho = 7$ and $\sigma_{\text{min}} = 0.002$ which corresponds to a negligible noise level when reaching the target distribution, $q$, assuming $\mathbb{V}[x_1]$ of order around 1. This method uses $N=18$ (NFE=35) steps to draw samples from the CIFAR-10 dataset, and $N=40$ (NFE=79) for ImageNet 64×64 and AFHQ 64×64.

**Sampling the LMM.** We use the sampling scheme used in (Song et al., 2023; Kim et al., 2024) to sample the LMM. This consists of the following iterations, $x^{t+1} = N_\theta(x^t, \sigma_t) + \sigma_{t+1}\eta$, where $x^0 \sim p_0$ and $\eta \sim \mathcal{N}(0, I)$. We report the noise scheduling we use in each step, $\sigma_t$, in terms of the ones used in (Karras et al., 2022), in Tables 3, 4, and 5.

**Training Cost.** The number of iterations used for training the LMM is listed in Table 6. The first 80k pre-training iterations were executed without the ADL as well as by evaluating the VGG-perceptual loss over 64-by-64 pixel images. This made each training iteration x6 faster than the following full-resolution and using the ADL. These numbers are lower than the ones reported in (Song et al., 2023), 800k for CIFAR-10 and 2400k for ImageNet 64×64, and in (Kim et al., 2024), 100k for CIFAR-10 and 120k for ImageNet 64×64. We note that these methods rely on having a pre-training DSM as in our case. Training the CFM (Lipman et al., 2023) does not require a pre-existing model, and uses 195k iterations for CIFAR-10 and 628k for ImageNet 64×64. The numbers of iterations quoted here are normalized to a batch size of 512.

| EDM | LMM | | | | EDM | LMM | | | |
|---|---|---|---|---|---|---|---|---|---|
| | wo/ADL | | w/ADL | | | wo/ADL | | w/ADL | |
| NFE 35 | NFE 1 | NFE 2 | NFE 1 | NFE 2 | NFE 79 | NFE 1 | NFE 2 | NFE 1 | NFE 2 |

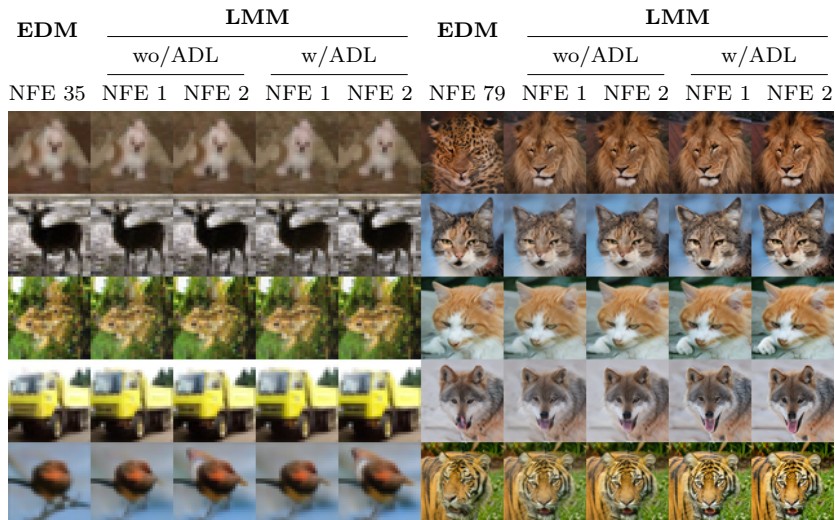

Table 7: CIFAR-10 (left) and AFHQ 64×64 (right) Samples Comparison.

Unlike the rest of these methods, the training data of the LMM must be first generated. As noted above, it consists of pairs of the form $x_0, N^*_{\text{Sampler}}(x_0)$ which are sampled from the EDM model $N^*$, in (Karras et al., 2022). The number of training examples we use for each dataset are listed in Table 6. On one hand this sampling process uses fairly high NFEs (35 for CIFAR-10, and 79 for ImageNet 64×64 and AFHQ 64×64), but on the other hand it consists of feed-forward executions with no back-propagation calculations. Moreover, this process can be executed on single GPUs and be trivially parallelized across multiple machines. In terms of wall-clock time this pre-processing did not take long, namely, half a day for CIFAR-10 compared to the 4 days of LMM training, and six days for ImageNet 64×64 compared to 20 days of training, and two

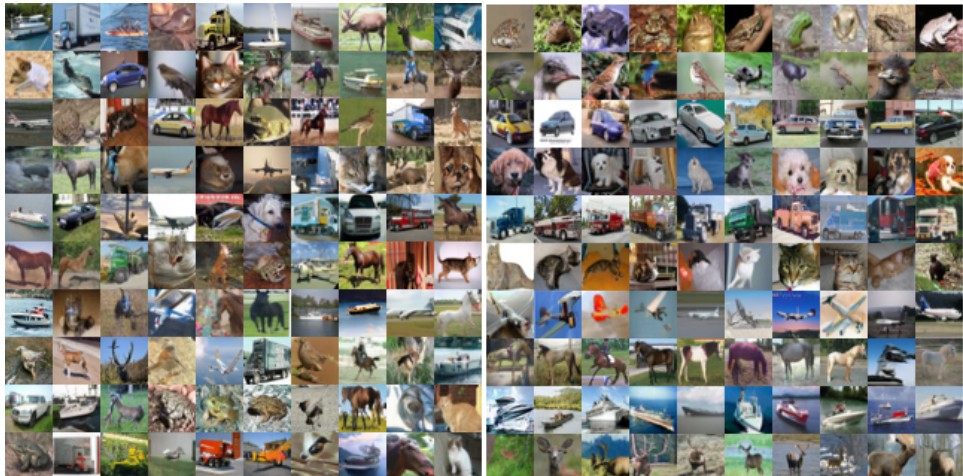

Figure 2: LMM Generated CIFAR-10 Samples. Class unconditional on the left, and conditional on the right. Rows correspond to different classes.

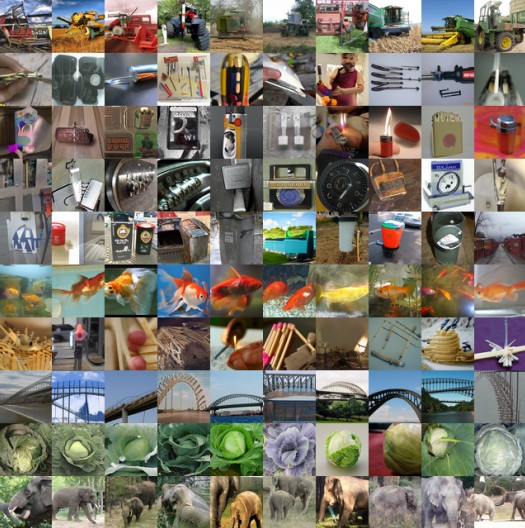

Figure 3: LMM Generated Conditional ImageNet 64×64 Samples. Rows correspond to different classes.

days for AFHQ 64×64 compared to 6 training days. We remind that these training sessions were conducted on four GeForce RTX 2080 Ti GPUs.

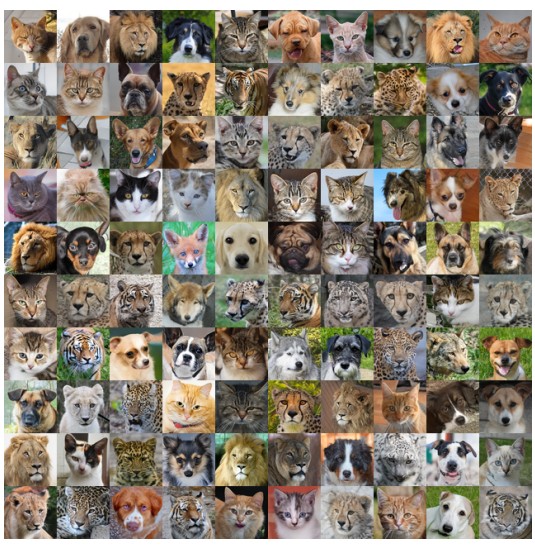

Figure 4: LMM Generated AFHQ 64×64 Samples.

