# OpenReview forum: "Denoising Efficiency and Lines Matching Models"
_TMLR — Rejected by TMLR_

### Review · Reviewer_LsYS · 2025-11-05

**Summary Of Contributions:**

The paper aims to develop an unconditional generative model capable of efficient sampling, particularly by promoting straightened generation trajectories. To this end, the authors propose a distillation-based approach that trains a denoiser to mimic linear interpolation trajectories between pairs of samples—from the prior to the data distribution—selected using a pre-trained diffusion model. Specifically, these initial–terminal pairs are obtained from the probability flow ODE (PF-ODE) solutions of the pre-trained diffusion model.

Formally, as encapsulated in Equation (7), given a prior sample $x_0$ and its PF-ODE solution $x_1$ (obtained from $x_0$), the proposed model is trained to predict $x_1$ from $x_1 + \sigma x_0$ across varying values of $\sigma$. Conceptually, this process resembles a single rectification step in ReFlow (Liu et al., 2022), but differs in two important aspects: (1) the terminal states are derived from a pre-trained diffusion model rather than the model being currently trained, and (2) the formulation directly models the denoised outcomes, akin to learning explicit PF-ODE solutions rather than estimating continuous-time velocity fields at each time step.

To further enhance sample quality, the paper adopts a perceptual loss in place of the standard mean-squared error and additionally incorporates an adversarial loss. Since the model is formulated as a denoiser, it naturally integrates with advanced ODE solvers proven effective for PF-ODEs, such as the second-order Heun’s method used in EDM (Kerras et al., 2022), which the authors adopt in their implementation.

In general, the proposed method is conceptually simple yet empirically promising. Its simplicity stems from several observations. First, as discussed in ReFlow++ (Lee et al., 2024), under mild assumptions, a single rectification step can be sufficient to achieve a straight generation path. Second, given that denoising diffusion can be reformulated within the flow-matching framework when a Gaussian distribution serves as the initial state, the use of PF-ODE aligns naturally with the idea of one-step rectification. In addition, the empirical improvements reported in the paper can be reasonably attributed to these design choices, making the overall approach intuitive and easy to interpret. The experiments presented in the paper demonstrate that the proposed model performs effectively within this framework.

Karras et al., Elucidating the Design Space of Diffusion-Based Generative Models, 2022
Liu et al., Flow Straight and Fast: Learning to Generate and Transfer Data with Rectified Flow 2022
Lee et al., Improving the Training of Rectified Flows, 2024

**Audience:**

No

**Audience Explanation:**

The paper addresses a timely and relevant topic in the generative modeling community—namely, the acceleration and simplification of sampling procedures in diffusion-based models through distillation. Given the current momentum surrounding diffusion model distillation, flow matching, and rectified generative trajectories, the general idea of employing PF-ODE–based supervision to guide a denoiser toward more efficient sampling is potentially of interest to TMLR’s readership.

However, there appears to be a gap between the paper’s stated claims and the presented empirical results. While the proposed approach exhibits some degree of novelty, its overall completeness and technical maturity seem limited. The empirical results are competitive, yet the absence of deeper theoretical grounding reduces the work’s overall impact. In its current form, certain parts of the paper could also be confusing to readers, as the narrative occasionally blurs the distinction between the method’s conceptual motivation and its empirical implementation.

In summary, while the paper may attract attention from researchers interested in diffusion model acceleration or generative path rectification, its broader appeal to the TMLR audience would likely depend on a stronger theoretical justification and clearer empirical evidence demonstrating advantages over established methods.

**Claims And Evidence:**

No

**Claims Explanation:**

The paper motivates its proposed method through the conceptual question of “induced correspondence vs. modeling the flow,” and presents a theoretical analysis that is claimed to be related to this distinction. However, the theoretical analysis provided lacks a clear and direct connection to the proposed method itself. Much of the discussion focuses on the limitations of existing flow-matching techniques, which are only tangentially relevant to the distillation formulation actually employed in this work. For instance, the analysis concerning the nonlinearity of generation paths under flow matching with conditional optimal transport (condOT) has already been extensively discussed in prior works such as *ReFlow* and many followup studies, yet the paper does not clearly explain how this analysis specifically motivates or supports the proposed approach. Similarly, the discussion regarding the limitations of batch OT has already appeared in earlier studies and does not provide new theoretical insights that are directly pertinent to the proposed framework. As a result, the theoretical discussion seems largely disconnected from the proposed method and provides limited support for its main technical claims.

A more meaningful theoretical contribution would have been to investigate whether, under the PF-ODE framework, a one-step rectification procedure inherently guarantees or approximates a linear correspondence between prior and data samples. Such an analysis would directly align with the paper’s central motivation of “straightening” the generative path and could clarify under what conditions PF-ODE trajectories preserve or deviate from linear interpolation in latent or data space. This line of inquiry is currently missing, leaving a conceptual gap between the stated theoretical motivation and the method’s actual formulation.

In addition, although the proposed approach appears empirically effective, it remains unclear to what extent the observed improvements can be directly attributed to the proposed method itself. In my understanding, the major components of the method can be categorized as follows: (1) the Equation (7) training formulation, (2) the use of EDM for the coupling in the rectification step, (3) the perceptual loss, and (4) the adversarial loss. Among these, Equation (7) constitutes the core novelty, yet the empirical analysis does not sufficiently isolate its contribution.

For example, in Tables 3–5, the ablation results suggest that perceptual and adversarial losses account for a substantial portion of the performance gains. However, it remains unclear how the model performs relative to other distillation methods when these auxiliary loss terms are removed. Admittedly, comparing absolute performance across baselines is not always straightforward, as each method may involve different degrees of tuning. Nonetheless, the paper would benefit from a clearer empirical analysis illustrating the specific effect of the proposed linear-interpolation-based formulation. In particular, showing how performance changes when perceptual and adversarial losses are excluded—while keeping the rest of the setup identical—would make the evidence for the paper’s core contribution much more convincing.

**Requested Changes:**

To improve the clarity, relevance, and overall impact of the work, the following revisions are recommended:

1. **Clarify the theoretical motivation and its connection to the proposed method.**

    The current theoretical section largely reiterates known results from the flow-matching and conditional optimal transport literature without explicitly linking them to the proposed PF-ODE–based distillation framework. The paper should make clear how the theoretical discussion informs or supports the proposed method, rather than presenting it as general background material.

2. **Strengthen the theoretical contribution.**

    A more valuable direction would be to examine whether one-step rectification under PF-ODE dynamics guarantees or approximates linear interpolation between prior and data samples. Such an analysis would directly support the paper’s central claim of “generation path straightening” and would help position the method as theoretically grounded rather than purely empirical.

3. **Provide a broader and more rigorous empirical comparison.**

    While the current experiments show promising results, they do not adequately clarify how much of the improvement can be attributed to the paper’s core contribution. Additional empirical analyses isolating the impact of each component—particularly the Equation (7) formulation—would allow readers to better assess the true advantages of the proposed approach relative to existing distillation methods.

---

> ### Author Response · Authors · 2025-11-18
> **Response to Reviewer LsYS**
>
> We thank the reviewer for his time and effort in reviewing our paper. The reviewer raised three concerns which he listed as requested changes. We address each of them:
>
> 1. Known observations and missing link between theory and experiments.
>
> We respectfully disagree with the first claim. As far as we know, and we will appreciate references from the reviewer if he knows otherwise, we are the first to:
>
> (i) show that the ill-posed diffusion loss at the limit of zero SNR creates a sample-independent basin of attraction to the probability flow lines causing them to curve - which in turn makes their integration harder (multiple sampling steps). The reviewer may have seen figures in papers showing these curved trajectories, and may have encountered works that mention the singular regression. But as far as we know - no one has linked the two.
>
> (ii) similarly, the required growth of batchsize in OT-based pairing was observed empirically in the literature (we cited an example Kim et al. (2023) in the context of generative models), but as far as we know we are the first to rigorously analyze this relation and provide a formal scaling relation with the signal dimension. If the reviewer is aware of such a proof, we will appreciate a reference. Furthermore, we note that methods that do use OT-pairing in the generative models community, namely,  [Tong 24] and [Poodlian 23], do not mention this scaling. Thus we believe our analysis is an important contribution to this field.
>
> Finally, the LMM is mainly used as a validation tool. Specifically, as our analysis pinpoints the lack of straightness of flow to the random pairing, the LMM shows this very aspect: random pairing is swapped with a deterministic one, and the straightness jumps. Moreover, the LMM pairs fully noised vectors with data points, i.e., it shows that the lack of straightness is not a matter of noise levels, but a matter of inconsistent pairing.
>
> In addition we discuss the question you raised about the ability to derive straight flows from straight paths (in the “Globally Straight Trajectories” subsection) where we raised the concern of line intersections. And again, the LMM served as a validation tool showing this limited extent of this concern (we further address this point below). We also note that the LMM straightness was also compared to OT based pairing at different batchsizes to substantiate the theoretical scaling with empirical validation in Table 1.
>
> While it is hard to get a perfect overlap between experimentation and theory, we believe the LMM served to support and substantiate several observations made.
>
> 2. Derive a theory on whether the endpoints of OF-ODE can be accurately approximated by linearized paths.
>
> Of course this is a most relevant and interesting question, especially in the context of the interpolation we, and other distillation approaches (RectFlow) use. First and foremost, we do not believe that a paper’s evaluation should be judged by bringing up other research topics, but based on the research topics they focus on.
>
> Secondly, while we agree that this is an interesting question, as we noted above the LMM served to assess the extent at which this linearization is a major bottleneck or not, and the answer it quite striking: given the paired examples the LMM is fully capable to reaching the teacher’s performance (Tables 3,4,5 in the appendix). Furthermore, with this linearization + an adversarial loss preventing distributional shift, the teacher’s performance is surpassed.
>
> Bottom line: we concluded that while this concern is relevant and interesting, it does not seem to be a major hurdle in practice given a fairly good teacher model. This was a reason for us to concentrate on other, more practically critical aspects to investigate.
>
> 3. Ablate the source of performance gain.
>
> As we noted the LMM is primarily used as a validation tool. Nevertheless, the combination of components “engineered” into it allowed it to reach SOTA results. We did not claim that it is the linear interpolation which achieves this, indeed we noted that without an adversarial loss, the LMM boils into an improved 2-RectFlow approach (+LPIPS) and our ablations show comparable results. By adding the adversarial loss, we avoid the distributional shift that the 2-RectFlow suffers from (and is discussed in the original paper), and only then, this new combination reaches the SOTA results.
>
> So the ablation study supports the claim that we make: it is NOT the linear interpolation which is responsible for the gain in quality, but its combination with the adversarial loss. Being an novel improvement of 2-RectFlow, the LMM provides a practical contribution on top of serving as a testbed for the theory. We will be happy to add additional testing to substantiate this claim if needed.
>
> We hope that these clarifications regarding the insights we make to central topics in diffusion models will allow the reviewer to reconsider the relevance fo our work to the TMLR community.

---

### Review · Reviewer_DB4m · 2025-11-05

**Summary Of Contributions:**

This paper provides analysis on the limitations of predominant diffusion model formulations, claiming they either collapse towards the mean of the data distribution or suffer from curse of dimensionality. The paper then provides an empirical exploration of a new flow model called LMM that regresses straight trajectories from a pretrained teacher diffusion model whose trajectory does not have to be straight. Experiments have shown improved metrics compared to other distilled models.

**Audience:**

Yes

**Audience Explanation:**

The paper has empirical value given the reported results compared to past methods.

**Broader Impact Concerns:**

No concern.

**Claims And Evidence:**

No

**Claims Explanation:**

First of all, the writing of the paper is extremely poor. Many notations are either undefined or overloaded in Sec 2. For instance, what are all the $p(x|x_1,t)$, $p(x^{t-1}|x^t)$, $p(x,\sigma_t)$? What exactly is being conditioned? They are all essentially the same (some sort of conditional Gaussian) but the notations are totally different. Beyond notation, the general writing is confusing and could use more polish to improve the clarity.

The paper focuses on distillation, yet the survey on related works focuses entirely on diffusion models, which are not the most relevant since the propsed method is based on having a pretrained diffusion model anyway.

The analysis part of the paper is not novel at all and the results are well known. For instance, (4) is the same as Karras 2022's Eq. (2,3), and the exponential scaling of batch size w.r.t. to the dimension is well-known in the statistical OT community.

The criticism about the optimal denoiser collapsing to the mean of the data distribution is unfounded. The presented interpretation is only correct in the extreme setting of large noise.

The proposed formulation (7) is not that different from directly regressing the teacher model's flow, except for the added noise. Since the experiment section focuses on 1-step or 2-step application of the proposed method anyway, it seems misleading to advertise the method as a flow model.

Although LMM performs well on the presented benchmarks, it is unclear to me how much of it is because of the auxiliary losses like the GAN loss, compared to the LMM formulation.

**Requested Changes:**

Please improve the clarity of the paper, shorten the discussion of less related works on diffusion models, and use better notations.

I would like to see more ablation studies to demonstrate the proposed method is actually a flow model, as opposed to just regressing a 1-step model.

---

> ### Author Response · Authors · 2025-11-17
> **Author's response to Reviewer DB4m**
>
> We thank the reviewer for his time in reading and commenting on our paper. It is hard for us to overlook the critical points raised, and feel we should first address them before going into the specific questions and requests.
>
> We believe the reviewer's comment “The paper focuses on distillation, yet the survey on related works focuses entirely on diffusion models, which are not the most relevant” may be the key to a misunderstanding.
>
> Our work is NOT about distillation, but it analyzes existing denoising approaches in an attempt to pinpoint the source of their inefficiency, later, when they are sampled.
>
> This is achieved by three separate contributions we make:
> 1. A theoretical analysis showing that the ambiguous pairing between latent source noise and target data samples leads to an ill-posed regression problem at the low SNR regimes. We link this singularity with a false attraction basin that curves the sampling trajectories, which in turn requires multiple sampling steps. We claim that linking this analytical singularity, which we show in three prominent models, to their efficiency at sampling time is the new insight our work offers.
>
> The reviewer seems to undermine this saying that it only happens in high noise levels - please note that EVERY SAMPLER STARTS at pure noise and hence suffers from this inefficiency - there is nothing negligible about it.
>
> 2. OT based pairing is perhaps the most well-known alternative strategy to the random pairing used in traditional DDPM/DSM models, and therefore we also theoretically investigated it. Out work provides a  rigorous proof showing that while this may be a valid approach for reducing the attraction to the false basins in simple problems, due to a fundamental course-of-dimensionality, the batch size required scales exponentially as a function of the signal dimension.
>
> The reviewer dismisses the value of having a formal proof by saying it is “well-known in the statistical OT community”. Two comments in this respect: (i) the papers proposing using this approach, e.g., for flow matching [Tong/Poodlian] do not mention this issue to the diffusion community. (ii) this striking scaling is of course hard to miss in practice, and indeed we cited a paper demonstrating it empirically [Kim et al. (2023)]. This however does not undermine the value of providing a rigorous analysis showing this scaling where it is quantified and its source is elucidated.
>
> 3. The LMM is primarily used for validating the sufficiency of our theoretical insights, and our goal is not developing yet another distillation model. Nevertheless, we still consider the LMM as a practical contribution given is favorable results on established benchmarks by “fixing” an issue in an existing Reflow model (namely, 2-Rectflow) where an adversarial loss is used to avoid the distributional drift from the data which RectFlow admits. We note that the contribution of several papers boils down to improving this popular method (e.g. by using LPIPS).
>
> We feel that the reviewer has either dismissed or ignored these contributions by saying “The analysis part of the paper is not novel at all and the results are well known. For instance, (4) is the same as Karras 2022's Eq. (2,3), and the exponential scaling of batch size w.r.t. to the dimension is well-known in the statistical OT community.”
> Eq. (4) is part of our background derivation - not a bottomline that we make - what’s wrong if it appears in Kerras? As we noted, the empirically observed scaling may be known - as we noted in the paper - but it was not rigorously investigated, and certainly not in the diffusion context of high signal dimension.
>
> Finally as to the required changes:
>
> The reviewer asks “What exactly is being conditioned?” in relation to p(x|x1,t), p(x^{t-1}|x^t,p(x,sig_t) and asks us to make this more clear. These are standard definitions of existing DDPM/DSM/FlowMatching models - basically copied and pasted to our background section. We believe it is clear that p(.) denotes a  probability density and p(.|.) a conditional one. We will make sure this is made clear in the paper as well as the fact that it is used to describe several models (which is the purpose of this section - to show the close relation between these models).
>
> The reviewer asks for ablations that show the LMM is indeed a flow model as opposed to regressing it as a 1 or 2 steps model. Tables 3,4 and 5 in Appendix C compare the results of a model trained over all possible times and the Sampling Optimized Training (SOT) approach where it is trained only on the specific times (which are later used at sampling). While the SOT shows some improvement, it is clear that even when trained as a flow model - at all times (noise levels) - the model outputs valid probability flow vectors also reaching competitive levels. If we failed to understand the reviewer's intent, we will be happy to hear and add experiments
>
> We thank again the reviewer for his effort and pled for his reconsideration.

---

### Review · Reviewer_3us6 · 2025-11-11

**Summary Of Contributions:**

The work presents the Lines Matching Model (LMM), a new probability flow model designed for highly efficient sampling. Its core idea is to first use a pre-trained, high-quality denoising model as a teacher to establish a deterministic mapping between noise vectors and data samples. Instead of distilling the teacher's potentially inefficient, curved sampling paths as done in prior work, LMM uses only these endpoint pairs to train a new model that follows globally straight lines. The method is further enhanced with perceptual and adversarial losses to achieve better quality. Its effectiveness is validated through experiments on established benchmark datasets.

**Audience:**

Yes

**Audience Explanation:**

- This work is well motivated by providing insightful theoretical analysis about weakness of random pairing in models and Optimal Transport (OT) solution.
- The idea of using endpoints to define an ideal path, rather than distilling an existing, imperfect path, is conceptually elegant and reasonable.

**Broader Impact Concerns:**

There is no ethical issue with this work.

**Claims And Evidence:**

Yes

**Claims Explanation:**

- This work is logically structured and well-written, and it provides a simple, 2D visualization figure that intuitively explains the core concepts and sound analysis that provides a strong theoretical underpinning.
- The proposed LMM method acheives SOTA performance on typical benchmark datasets.

**Requested Changes:**

- The proposed algorithm is built on top of a powerful, pre-trained ODE-based teacher model like EDM to produce the initial pairings. This makes the overall process a two-stage pipeline, which is more complex and time-consuming than methods that train from scratch.
The data manifold of natural images is highly non-linear. It's theoretically possible for a linear path to traverse very low-density regions, potentially leading to unnatural intermediate states or artifacts. Could authors provide any explanation about this?
- Adversarial loss and perceptual loss are very common in generative model training. They could be easily equipped to other flow matching models and has been employed in SiD. Hence, the novelty here is limited.
- In addition, like many GAN-hybrid models, performance may be sensitive to the specifics of the adversarial training setup, which can require careful tuning.
- The results show that another one-step method, Score Identity Distillation (SiD), is highly competitive and even outperforms LMM at NFE=1 on AFHQ 64x64. Authors should provide analysis and explanation about this reults.

---

> ### Author Response · Authors · 2025-11-17
> **Author response to Reviewer 3us6**
>
> We thank the reviewer for his time in reading and commenting on our paper, as well as appreciating the contributions made.
>
> Let us jump to the changes requested:
>
> 1. “ The data manifold of natural images is highly non-linear. It's theoretically possible for a linear path to traverse very low-density regions, potentially leading to unnatural intermediate states or artifacts. Could authors provide any explanation about this?
>
> Of course, this is a very natural question to ask. Every diffusion model ends up inducing a pairing between the source (noise) and target (data) distributions. This means that these endpoints can be tied by the lines connecting them - no problem so far.
> The problem that may occur is that these lines may intersect - meaning that the flow field they define is undermined. The closer they are to lines in the teacher model, the less likely this would happen (so EDM is a good choice).
> We discussed this issue in the paper under the subsection titled “Globally Straight Trajectories” on page 7. Given your question - we will clarify that this discussion is also related to the more basic question of why global lines are something to be expected.
>
> 2. The use of the adversarial loss is two fold: (i) it provides a better norm to optimize the model with - and models that can accommodate this loss (e.g. CTM) use it. The LMM can also accommodate it so for a fair comparison we also used it. We agree that there is nothing novel about that. However, (ii) its use in the LMM boils down to augmenting the 2-RectFlow with a loss that ties this iterative model to the data (originally this method departs from the data in its additional iterations). This solves a distributional shift that RectFlow’s paper admitted. In this respect - the LMM offers a non-trivial solution to a well-accepted model, which we hope you can see novelty in. This is discussed in the paper at the end of Section 2 (page 4).
>
> 3. You are obviously correct, GANs are greatly affected by parameter tuning. At the appendix, we noted in the paper that we use the models and parameters used in [Sauer et al., 2022, Esser et al 2021] AS IS without further tuning.
>
> 4. Indeed as you are correct again, SiD is a strong competitor outperforming the LMM on one one test - the AFHQ, NFE=1. We actually discussed the reason why this happens at the end of Section 4 by saying “While achieving a state-of-the-art FID of 1.54 at NFE=2, the SiD achieves a better score using a single step. We note that unlike the CIFAR-10 and ImageNet 64×64 cases, the discriminator architecture and hyper-parameters we used were not we used were not tailored to this dataset in previous work (e.g., StyleGAN-XL (Sauer et al., 2022)). This affected the expected improvements from the SOT strategy, as discussed in Appendix C, and we therefore believe the LMM has greater potential on this dataset.”
>
> Please let us know if the explanations and changes suggested here are sufficient in your opinion. We thank again the reviewer for his time.

---

### Decision · Action_Editor_cfaH · 2026-01-31

**Recommendation:** Reject

**Additional Comments:**

While the paper proposes an interesting method that achieves compelling results on established benchmarks, the reviewers’ concerns do not appear to have been sufficiently addressed in the current revision. Therefore, a major revision is encouraged to address the following points:

1. Positioning: the paper claims to identify the "root of the sampling inefficiency of modern diffusion models". However, reviewers noted that the core arguments (i.e., mean predictions in high noise levels and the importance of couplings) are not fundamentally novel. Given the reliance on a pre-trained models and the connections to existing methods (see below), the work should be positioned as an improvement over current distillation methods.
2. Contextualization:
    - The paper identifies mean regression in denoising high noise levels as an "ill-posed" problem, but reviewers noted that predicting the conditional mean is a well-known property.
    - While similarities to rectified flows (and extensions such as RF++ and simple ReFlow) are acknowledged, reviewers noted the methods are more closely related than the paper specifies. Both utilize deterministic pairings to learn models with straighter paths for efficient sampling and the differences between 2-Rect-Flow (with VE schedule) and LMM are not clearly discussed in the paper. The claim that 2-Rect-Flow "lacks the ability" to use domain-specific losses can be challenged since this is merely a matter of parameterization (flow-prediction vs. $x_{0}$-prediction).
3. Experiments: Reviewers asked for further experiments to ablate the improvements of different components of LMM over baselines. The paper acknowledges the importance of perceptual and adversarial losses (and motivates them by their ability to mitigate distribution shifts in reflow). However, it should also be explicitly indicated which methods utilize external models and whether these have been pre-trained on additional data.

**Audience:**

Yes

**Audience Explanation:**

The findings would be relevant to researchers working on diffusion/flow models and distillation.

**Claims And Evidence:**

No

**Claims Explanation:**

See additional comments below.

**Resubmission Of Major Revision:**

The authors may consider submitting a major revision at a later time.